# SonoGym: High Performance Simulation for Challenging Surgical Tasks with Robotic Ultrasound

**Yunke Ao**
Balgrist University Hospital
ETH Zurich

**Masoud Moghani** *
University of Toronto
NVIDIA

**Mayank Mittal** *
ETH Zurich
NVIDIA

**Manish Prajapat**
ETH AI Center
ETH Zurich

**Luohong Wu**
Balgrist University Hospital
University of Zurich

**Frederic Giraud**
Balgrist University Hospital
University of Zurich

**Fabio Carrillo**
Balgrist University Hospital
University of Zurich

**Andreas Krause**
ETH AI Center
ETH Zurich

**Philipp Fürnstahl**
Balgrist University Hospital
University of Zurich

## Abstract

Ultrasound (US) is a widely used medical imaging modality due to its real-time capabilities, non-invasive nature, and cost-effectiveness. Robotic ultrasound can further enhance its utility by reducing operator dependence and improving access to complex anatomical regions. For this, while deep reinforcement learning (DRL) and imitation learning (IL) have shown potential for autonomous navigation, their use in complex surgical tasks such as anatomy reconstruction and surgical guidance remains limited — largely due to the lack of realistic and efficient simulation environments tailored to these tasks. We introduce SonoGym, a scalable simulation platform for complex robotic ultrasound tasks that enables parallel simulation across tens to hundreds of environments. Our framework supports realistic and real-time simulation of US data from CT-derived 3D models of the anatomy through both a physics-based and a generative modeling approach. Sonogym enables the training of DRL and recent IL agents (vision transformers and diffusion policies) for relevant tasks in robotic orthopedic surgery by integrating common robotic platforms and orthopedic end effectors. We further incorporate submodular DRL—a recent method that handles history-dependent rewards—for anatomy reconstruction and safe reinforcement learning for surgery. Our results demonstrate successful policy learning across a range of scenarios, while also highlighting the limitations of current methods in clinically relevant environments. We believe our simulation can facilitate research in robot learning approaches for such challenging robotic surgery applications. Dataset, codes, and videos are publicly available at `https://sonogym.github.io/`.

## 1 Introduction

Ultrasound is a commonly used medical imaging technique because it is non-invasive, cost-effective, and capable of providing real-time images [19]. While freehand ultrasound is operator-dependent and skill-intensive, *robotic ultrasound* systems have been developed to enhance reproducibility and

---

*Equal second author contribution. Correspondance to: `yunke.ao@balgrist.ch`

39th Conference on Neural Information Processing Systems (NeurIPS 2025) Track on Datasets and Benchmarks.

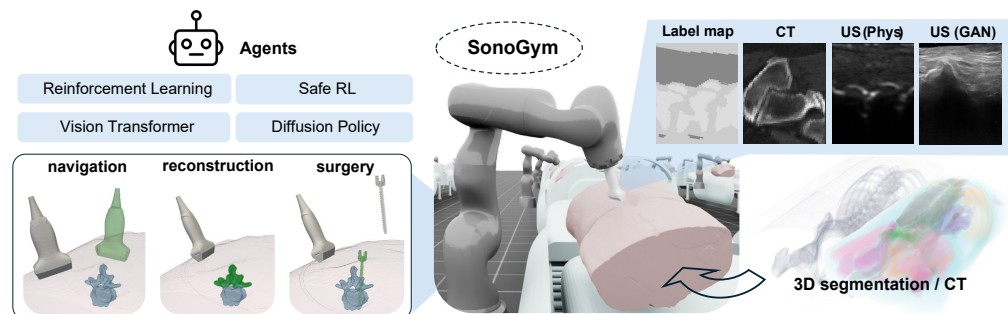

Figure 1: **Overview.** SonoGym provides model-based and learning-based ultrasound (US) simulation using 3D label map and CT scans from real patient datasets. Tasks in SonoGym include US navigation, anatomy reconstruction, and US-guided robotic surgery. SonoGym enables benchmarking of various algorithms, including reinforcement learning (RL), safe RL, vision transformer, and diffusion policy.

improve imaging efficiency [20]. Additionally, due to its ability to penetrate soft tissues, robotic ultrasound has been used for intraoperative guidance in various surgical procedures [39].

Robotic ultrasound has been particularly impactful and widely adopted in orthopedic surgery, which often has limited visibility and high precision requirements. These applications can be categorized into three main tasks: *navigation*, anatomy *reconstruction*, and ultrasound-guided *surgery*. For *navigation*, the ultrasound probe can be manipulated autonomously to localize and track target anatomy [54, 6]. For anatomy *reconstruction*, robotic ultrasound has been used to scan and reconstruct the dorsal surface of the spine, which can then be registered with preoperative CT images to guide surgical steps [27, 52]. While these systems employ *heuristic* path planning for the ultrasound probe, determining an *optimal* scanning path based solely on ultrasound image feedback remains a challenging open problem. Complete robotic ultrasound-guided spinal *surgery* pipelines have also been developed [25, 28], but they often rely on registration between ultrasound and preoperative CT images, lacking more intelligent planning directly informed by ultrasound image inputs.

Deep Reinforcement Learning (DRL) and Imitation Learning (IL) have shown strong potential in addressing such complex vision-based decision-making problems [21, 33, 7, 60]. Numerous studies have applied DRL and IL to autonomous robotic ultrasound *navigation* [15, 37, 26]. However, their use in ultrasound-guided *reconstruction* and orthopedic *surgery* remains largely unexplored.

One key limitation to the broader use of DRL and IL in robotic ultrasound is the absence of high-performance, realistic simulation environments. In other areas of robotics, simulation-based DRL training has proven highly effective [33, 1, 17, 50], supported by platforms such as NVIDIA IsaacLab [34], IsaacGym [31], PyBullet [8], and Mujoco [51]. In this work, our objective is to develop a comprehensive, efficient robotic ultrasound simulation platform that enables simulating not only *navigation*, but also other challenging tasks covered by ultrasound-guided surgical procedures, such as *reconstruction* and execution of *surgery*.

Our main contributions are summarized as follows: (i) We present a realistic and efficient robotic ultrasound simulation platform, SonoGym, which includes multiple anatomical models of the real patient from TotalSegmentator [55] and supports both model-based and learning-based ultrasound simulation, as shown in Fig.1. (ii) We formulate ultrasound-guided navigation, reconstruction, and surgery as specialized Markov Decision Processes (MDPs), enabling the training of high-performing DRL agents. To more effectively capture task-specific challenges, we adapt these models to partially observable MDPs (POMDPs), submodular MDPs [40], and state-wise constrained MDPs [61]. (iii) We generate expert demonstration datasets within our simulator to enable training of recent IL agents, including the Action Chunking Transformer (ACT, [60]) and the Diffusion Policy (DP, [7]). (iv) We conduct extensive evaluations and comparisons of DRL and IL approaches across the different tasks, analyzing their generalization performance across different ultrasound noise and different patient models; These results help assess the potential of DRL and IL and the challenges that remain in this application domain.

## 2 Related Works

**Simulation platforms for robotic surgery**. Multiple simulation platforms have been developed to support DRL training for various surgical tasks. LapGym [42] provides a simulation environment for robot-assisted laparoscopic surgery with soft-tissue deformation based on the SOFA framework [11]. SurRol [57, 30] enables surgical robot learning compatible with the da Vinci Research Kit [23], based on PyBullet. Surgical Gym [43] allows the training of various surgical robotic arms to reach desired positions. Orbit-surgical [36, 58] provides various surgical manipulation environments with photorealistic rendering, enabling the training of visuomotor (from vision to action) policies using DRL or IL. So far, most existing surgical simulation platforms focus on laparoscopic (minimally invasive) surgery or soft tissue manipulation, with few providing realistic patient models like [36] and intraoperative medical imaging modalities such as ultrasound.

**Efficient ultrasound simulation**. Real-time and realistic ultrasound simulation has been a long-standing research topic. Traditional approaches employ GPU-accelerated ray tracing based on CT images or segmentation maps [24, 41, 46, 4, 10, 32]. We adopt a model-based simulation approach based on [41, 24] in our framework. Recently, generative networks have also been leveraged for ultrasound image simulation, enabling more realistic image patterns while maintaining fast inference. For example, Liang *et al.* [29] and Alsinan *et al.* [2] explored the generation of ultrasound images based on composite label maps or bone sketches using Generative Adversarial Networks (GAN, [14]). Song *et al.* [48] studied learning-based CT-to-ultrasound translation with CycleGAN [62]. However, these approaches have not been utilized for training DRL or IL agents. Although diverse and realistic ultrasound images can also be generated using diffusion models [49, 9, 22], we adopt a GAN-based approach [18] to maintain simulation efficiency. With a high-quality in-house paired CT-ultrasound dataset [56], we can train effective GANs tailored for orthopedic surgery.

**Simulation and robot learning for robotic ultrasound**. DRL or IL-based robotic ultrasound navigation has been widely explored. However, many existing works utilize their own ultrasound sweep dataset or simulation. For instance, Hase *et al.* [15] and Li *et al.* [26] applied deep Q-learning on in-house collected ultrasound sweeps to learn a navigation policy for the spine. Ning *et al.* [37] develop a robotic ultrasound environment to train a policy for ultrasound image acquisition based on RGB image observation, therefore, their simulation only involves the RGB camera, robot and soft tissues, without incorporating an ultrasound simulation. Ao *et al.* [3] investigate intraoperative surgical planning based on reconstructed bone surface from ultrasound, but they only simulate noisy bone surface reconstructions without raw ultrasound images. In contrast to them, our work further provides realistic ultrasound image simulation for orthopedic surgery and relevant anatomy, along with incorporating challenging tasks such as ultrasound-guided *surgery* and bone surface *reconstruction*.

## 3 Preliminaries

### 3.1 Reinforcement learning

**Markov decision process**. An MDP is a tuple of $\langle \mathcal{S}, \mathcal{A}, \mathcal{P}, \rho, \mathcal{O}, T, \mathcal{R} \rangle$, where $\mathcal{S}$ is the state space with state $s \in \mathcal{S}$, $\mathcal{A}$ is the action space with action $a \in \mathcal{A}$, $\mathcal{P}$ is the transition probability, $\rho$ is the initial state distribution, $\mathcal{O}$ is the observation space with observation $o \in \mathcal{O}$, and $\mathcal{R}$ is the reward space with $r \in \mathcal{R}$. An episode starts at $s_0 \sim \rho$, and at each time step $t \geq 0$ at state $s_t$, the agent receives an observation $o_t$ and it draws an action $a_t$ conditioned on it according to a policy $\pi : \mathcal{O} \times \mathcal{A} \to [0, 1]$. Applying this action, the simulation environment evolves to a new state $s_{t+1}$ following the MDP transition $\mathcal{P}$ and receives a reward $r(s, a)$. For the typical RL task described above, we deploy the proximal policy optimization (PPO, [44]) and Advantage Actor Critic (A2C, [35]) algorithms.

**State-wise constrained MDP**. The MDP is appended with the cost functions $C_i : \mathcal{S} \times \mathcal{A} \to \mathbb{R}, \forall i \in [m]$ with a total of $m$ constraints [61]. The feasible policy class $\Pi_c$ for this MDP satisfies $\mathbb{E}[C_i(s_t, a_t)] \leq w_i, \forall i \in [m]$ and $\forall t \in [T]$ where $w_i$ are constraint thresholds. For the state-wise constrained RL problem, we implement a modified version of SafeRPlan [3], in which the safety distance prediction network is adapted to predict the cost $\hat{C}_i(s_t, a_t)$ instead of the distance. During testing, no action is taken if any cost prediction exceeds a threshold, *i.e.*, $\hat{C}_i(s_t, a_t) > w_i - \delta$, where $\delta$ is a margin that adjusts the level of conservatism.

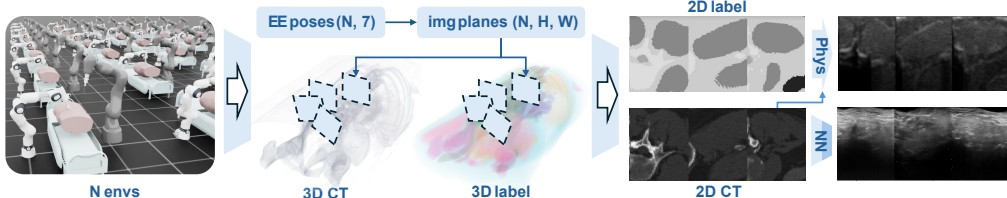

Figure 2: **Efficient ultrasound simulation across a large number of environments.** Given the current end-effector poses of the robot arms, we first compute the ultrasound image planes in the patient frames attached with 3D CT volumes and label maps. We then extract 2D CT and label slices as pixels on the plane. Ultrasound images are subsequently simulated based on these inputs using either physics-based models or neural networks.

**Submodular MDP**. The MDP here is appended with non-additive trajectory-based rewards, in contrast to state rewards defined above. We replace $\mathcal{R}$ with a set function $F : 2^{T \times S} \to \mathbb{R}$, which is a monotone submodular function (*c.f.* [40]). We denote the trajectory with $\tau$ and for each (partial) trajectory $\tau_{l:l'}$, we use the notation $F(\tau_{l:l'})$ to refer to the objective $F$ evaluated on the set of (state, time)-pairs visited by $\tau_{l:l'}$. Computationally, solving the submodular MDP problem is intractable; however as proposed in [40], we use a PPO variant of submodular policy optimization, *i.e.*, use marginal gain as reward at state $r(s_t, a_t) = F(s_{t+1}|\tau_{0:t}) := F(\tau_{0:t+1}) - F(\tau_{0:t})$. This results in greedy maximization of rewards at each step, which is empirically shown to perform well [40].

## 3.2 Imitation learning

**Action chunking transformer [60]**. In ACT, a policy is learned to predict a sequence of actions (action chunk) given the current observation $\pi(a_{t:t+k}|o_t)$, where $k$ is the sequence length. To address the noise from the expert data, the policy is trained as a conditional variational autoencoder (CVAE, [47]), using Transformer architectures [53]. During inference, the policy predict an action chunk at each time step, and the final action is smoothed using a *temporal ensemble* (averaging predictions from different previous steps).

**Diffusion policy (DP) [7]**. In DP, a visuomotor policy $\pi_\theta(a_{t:t+k}|o_{t-h:t})$ is trained using Denoising Diffusion Probabilistic Models (DDPM, [16]) to predict a sequence of actions $A_t := a_{t:t+k}$ given the historical visual observation input $O_t := o_{t-h:t}$, where $k$ is the horizon length, $h$ is the number of steps for the latest observations. Then DDPM performs $M$ denoising steps to generate $A_t^{M-1}, A_t^{M-2}, ..., A_t^0$ based on a noise prediction network $\epsilon_\theta(O_t, A_t^m, m)$:

$$A_t^{m-1} = \alpha(A_t^m - \gamma\epsilon_\theta(O_t, A_t^m, m) + \eta_m), \quad m = M, M-1, ..., 1, \quad \eta_m \sim \mathcal{N}(0, \sigma^2 I),$$

where $\alpha, \gamma, \sigma$ are determined by the noise scheduler. $\epsilon_\theta$ is trained using pairs of Gaussian noise $\eta_m$ and ground truth actions $\bar{A}_t^0$ by minimizing loss $\|\eta_m - \epsilon_\theta(O_t, \bar{A}_t^0 + \beta_m\eta_m, m)\|^2$, where $\beta_m$ depends on the noise scheduler.

## 4 SonoGym Environments

We discuss the three robotic ultrasound-guided tasks: *navigation*, bone surface *reconstruction*, and spinal *surgery*, supported by our SonoGym platform with realistic ultrasound simulations, as is shown in Fig.1. In the following, we introduce the components and tasks of SonoGym in detail.

**Assets**. We describe the main components of our simulation environments: robot arms, patient models, and end effectors (surgical drills and ultrasound probes). We support multiple robotic arm models, including KUKA Med14 and Franka Emika Panda, both of which can be equipped with either an ultrasound probe or a drill at the end effector. The robot arm simulation and its corresponding inverse kinematic controllers are adopted from NVIDIA IsaacLab. We provide patient models derived from the TotalSegmentator dataset [55], which include 3D anatomical segmentations and corresponding CT images. For each patient in a subset of the dataset, we generate target trajectories on the L4 vertebra to guide robotic drilling, which serves as the objective for the *surgery* task.

**Ultrasound simulation**. We explain the pipeline to simulate ultrasound images based on patient-specific 3D CT scans and segmentation maps, as shown in Fig.2. We first compute the ultrasound image planes in the patient coordinate frames, using the end-effector poses of the robot arm. We then extract the corresponding pixels on the image planes from the 3D CT volume and segmentation map to generate 2D CT and label slices. For model-based (MB) simulation, we adopt the convolution-based method from [41] to simulate ultrasound images from the label slice, and refine the reflection term using the CT slice similar to [24]. For learning-based (LB) simulation, we train a generative model using the pix2pix framework [18] to translate CT slices into ultrasound images, leveraging a large in-house CT-to-ultrasound paired dataset collected from 7 ex-vivo spine specimens [56]. We apply intensity histogram matching between the input CT slices and a subset of training CT images to mitigate the domain gap in learning-based simulation. Both learning-based and model-based simulations are executed in batch mode across parallel environments to maintain computational efficiency.

## 4.1 Task 1: Ultrasound navigation

**Task description**. We consider the problem of navigating an ultrasound probe to locate a target anatomy, starting from a random initial position on the back of the patient. The target pose of the ultrasound probe with frame $\{U\}$ is represented by a fixed goal frame $\{G\}$ located above the target anatomy, shown in orange in Fig.3 (left). In practice, the precise frame of the target anatomy $\{G\}$ is typically *unknown* in a real patient. However, the real-time ultrasound image partially visualizes the underlying anatomy and implicitly encodes the position of the probe ($\{U\}$) relative to the goal frame ($\{G\}$), which can be exploited for navigation. To ensure continuous image acquisition, the ultrasound probe must maintain stable contact with the skin. We assume this contact is maintained by an existing low-level robot controller, which also enforces the probe to remain perpendicular to the skin surface (*i.e.*, the $z$-axis of the probe aligns with the local surface normal). Our focus is solely on task-space planning of the *3 DoF tangential motion* of the ultrasound probe along the skin surface to reach the goal frame.

**States and observations**. At any time $t$, the state $s_t \in \mathcal{S} \subseteq \mathbb{R}^6$ consists of the relative position $_U^G p_t \in \mathbb{R}^3$ and angle-axis orientation $_U^G q_t \in \mathbb{R}^3$ between the ultrasound probe frame $U$ and the goal frame $G$. The observation is the real-time ultrasound image feedback $o_t \in \mathcal{O} \subseteq \mathbb{R}^{H \times W}$, where $H, W$ denote the height and width of the images, respectively.

**Actions and reward**. The action is defined over the remaining degrees of freedom in the ultrasound probe frame $U$ as $a_t := [\Delta x_t, \Delta y_t, \Delta \alpha_t]^\top \in \mathbb{R}^3$, where $[\Delta x_t, \Delta y_t]^\top$ represents horizontal translations on the skin surface (along the $x$ and $y$ axes of $U$, shown green in Fig. 3 (left)), and $\Delta \alpha_t$ denotes rotation around the surface normal ($z$ axis of $\{U\}$). The reward is defined as the change in distance to the goal frame: $r_t = w_1(\|_U^G p_t\| - \|_U^G p_{t+1}\|) + \|_U^G q_t\| - \|_U^G q_{t+1}\|$, where $w_1$ is a tunable weight.

**Agents**. We support training PPO agents, which achieve high performance for navigation. For the network architecture, we use a shared convolutional neural network (CNN) encoder for the policy and value networks. Furthermore, we provide datasets for the training of imitation learning agents (such as ACT and DP), which are collected with an *expert policy* based on the true state $_U^G p_t$ and $_U^G q_t$. The *expert policy* is defined as $a_t^* = \rho_1[_U^G p_t \cdot e_x, _U^G p_t \cdot e_y, _U^G q_{t+1} \cdot e_z]^\top$, where $\rho_1 < 1$ is the proportional scaling parameter, $e_x, e_y, e_z$ denotes the unit vectors along the $x, y, z$ axes of $\{U\}$. We train ACT and DP with the default architecture described in [60, 7].

## 4.2 Task 2: Bone surface reconstruction

**Task description**. We consider the robotic ultrasound spine surface reconstruction problem following the setup in [27], and simplify the task to reconstruct only the surface of a single vertebra. We assume that the bone surface can be segmented from each 2D ultrasound image using segmentation methods such as [38]. The 3D reconstruction of the vertebra surface can be obtained by combining these segmentations across frames, together with the tracked pose of ultrasound probes. Our task focuses on optimal path planning of the ultrasound probe on the skin surface to enable fast and sufficient reconstruction for registration, as is shown in Fig. 3 (middle). Compared to the navigation task setting, we additionally allow adjustment of the pitch angle (rotation around the $y$-axis of the $\{U\}$ frame) to capture more surface points. This planning problem is challenging because the pose of the target

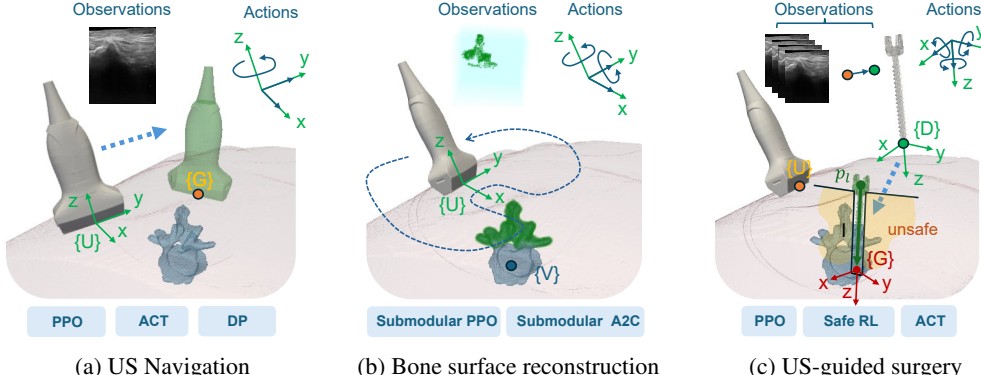

| PPO | ACT | DP |
|-----|-----|-----|

| Submodular PPO | Submodular A2C |
|----------------|----------------|

| PPO | Safe RL | ACT |
|-----|---------|-----|

(a) US Navigation          (b) Bone surface reconstruction          (c) US-guided surgery

Figure 3: **Tasks.** The target anatomy (L4 vertebra) is colored dark blue. **(a) Ultrasound (US) navigation:** Move the ultrasound probe to the goal pose (green) based on the real-time ultrasound images. **(b) Bone surface reconstruction:** Efficiently scan the surface of the target vertebra (green) with a low path length. **(c) Ultrasound-guided spinal surgery:** Fix the ultrasound probe to track the target vertebra, and drill inside the vertebra safely based on the ultrasound image and tracked poses of the ultrasound probe and drill. Supported algorithms are illustrated below each task.

vertebra $\{V\}$ is unknown, requiring the planner to balance exploration and exploitation based on the current reconstruction.

**State and observations.** Our state $s_t$ is the (unknown) position $_V^U p_t \in \mathbb{R}^3$ and angle-axis orientation $_V^U q_t \in \mathbb{R}^3$ of the ultrasound probe in the target vertebra frame $\{V\}$. The observation is defined as the current reconstruction $\mathcal{M}_{0:t}$ transformed to the current ultrasound frame $_U\mathcal{M}_{0:t}$ and voxelized to a 3D image of shape $H \times W \times E$, as is shown in Fig.3 (middle), where $H, W, E$ denote the image height, width, and elevation, respectively. It partially encodes the current reconstruction status and the relative poses $_V^U p_t, _V^U q_t$.

**Actions and reward.** The 4D action is defined as $a_t := [\Delta x_t, \Delta y_t, \Delta \alpha_t, \Delta \beta_t]^\top$, corresponding to translation along $x, y$ axes and rotation around $z, y$ axes in $\{U\}$ frame. The trajectory objective function $F$ accounts for both the total number of acquired surface points and the trajectory length, and is given by: $F(\tau_{0:t}) := |\mathcal{M}_{0:t}| - w_2 \sum_{h=0}^{t}(|\Delta x_h| + |\Delta y_h| + w_3|\Delta \alpha_h| + w_3|\Delta \beta_h|)$, where $w_2, w_3$ are tunable weights, $|\mathcal{M}_{0:t}|$ denotes the area of the surface $\mathcal{M}_{0:t}$.

**Agents.** We support training submodular PPO and A2C, which achieved strong performance in our task. Both policy and value function networks share a CNN encoder, with separate heads for actions and values. For comparison, we also provide heuristic open-loop path planning, following the approach described in [27].

### 4.3    Task3: Ultrasound-guided surgery

**Task description.** We consider the ultrasound-guided path planning problem in robotic bone drilling for pedicle screw placement. During surgery on a real patient, the exact position of the target vertebra to be drilled is not directly known. To localize the target vertebra, we assume a 3D ultrasound probe is navigated above the region of interest, as shown in Fig.3 (right), continuously acquiring volumetric ultrasound images $I_t \in \mathbb{R}^{H \times W \times E}$. The poses of both the ultrasound probe frame $\{U\}$ and the drill frame $\{D\}$ are tracked by the robotic tracking system. The objective is to safely and accurately drill into the vertebra by following a predefined path (green in Fig. 3 (right)), using the acquired ultrasound images and pose tracking. This path starts from a point $p_l$ on the skin surface and leads to a target goal frame ($\{G\}$, red), which is defined by the surgeon directly on the vertebra.

**States and observations.** The state $s_t \in \mathcal{S} \subseteq \mathbb{R}^6$ consists of the relative position $_G^D p_t \in \mathbb{R}^3$ and angle-axis orientation $_G^D q_t \in \mathbb{R}^3$ between the goal frame $\{G\}$ and the current drill frame $\{D\}$. The observation $o_t$ includes the volumetric ultrasound images $I_t$ and the tracked relative pose between the ultrasound probe and the drill, $[_U^G p_t, _U^G q'_t] \in \mathbb{R}^7$, where $_U^G q'_t$ is a quaternion. To simulate noise in ultrasound-based navigation, we randomize the position of the ultrasound probe above the target vertebra by up to a user-specified threshold $\lambda$.

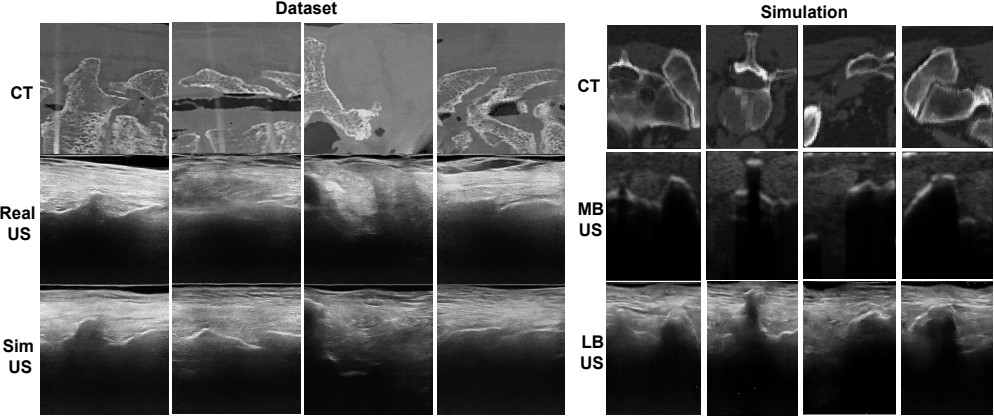

Figure 4: **Qualitative evaluation of ultrasound (US) simulation.** (left) Evaluation with the testing dataset. The top row shows 2D CT slices of the human specimen. The corresponding real US and the generated US of Sonogym are shown in the rows below. (right) Examples of simulated US images in SonoGym. Sliced CT, model-based (MB) US simulation, and learning-based US simulation are shown in the top, middle, and bottom rows, respectively. The learning-based US simulation has high visual similarity to real US images.

**Actions, reward and cost**. The action is defined as the 6D command for the drill in $\{D\}$ frame: $a_t := [_D\Delta p_t, {}_D\Delta q_t]$, where $_D\Delta p_t$ is the translation and $_D\Delta q_t$ is the rotation vector. While the drill can move freely outside the patient, it must remain within a narrow region once inside to ensure safety, as illustrated in Fig. 3 (right). To facilitate reward design, we define the free region $\mathcal{C}_{\text{free}}$ and drilling region $\mathcal{C}_{\text{drill}}$ in $\{G\}$ as follows:

$$\mathcal{C}_{\text{drill}} := \{p \mid \sqrt{p_x^2 + p_y^2} \leq \frac{d}{2}, \ -l \leq p_z \leq 0\}, \quad \mathcal{C}_{\text{free}} := \{p \mid p_z \leq -l\},$$

where $p_x, p_y, p_z$ are the $x, y, z$ components of the arbitrary position $p$, $l$ is the distance from skin to the goal, $d$ is the diameter of the drill region. The remaining space is defined as the unsafe region $\mathcal{C}_{\text{unsafe}}$. The reward is defined as

$$r_t := \begin{cases} w_4 \left( \|_G^D p_t - p_l\| - \|_G^D p_{t+1} - p_l\| \right) + w_5 \left( \|_G^D q_t\| - \|_G^D q_{t+1}\| \right), & \text{if } _G^D p_t \in \mathcal{C}_{free}, \\ w_6 \left( \|_G^D p_t\| - \|_G^D p_{t+1}\| \right) + w_5 \left( \|_G^D q_t\| - \|_G^D q_{t+1}\| \right), & \text{if } _G^D p_t \in \mathcal{C}_{drill}, \end{cases}$$

and 0 otherwise. Here $p_l := [0, 0, l]$ is the surface point directly above the goal, $w_4, w_5, w_6$ are tunable weights. This reward encourages agents to first reach the skin point $p_l$ before beginning the drilling process. The cost for the state-wise constrained MDP is the indicator function of the unsafe region $c_t := \mathbb{I}\left[_G^D p_t \in \mathcal{C}_{unsafe}\right]$.

**Agents**. We provide both PPO and modified SafeRPlan agents for the task. A CNN and an MLP are used as encoders for the image and relative pose observations, respectively. We also provide datasets collected from an expert policy (with expert action $a_t^*$), which moves the drill towards the skin point $p_l$ first, then towards $\{G\}$. We support the training of ACT using the collected dataset.

## 5 Experiments

In this section, we demonstrate the effectiveness of the proposed simulation environment by training and comparing the performance of RL and IL algorithms. In the experimental study, we are interested in answering the following four research questions. (1) How realistic and efficient is the ultrasound simulation? (2) How effective are MDP formulations and reward design for different tasks? (3) How do RL and IL compare in performance? (4) Can pre-training on multiple ultrasound simulation models and patients enable zero-shot generalization to unseen ultrasound noise and patients?

**Metrics**. We quantitatively evaluated ultrasound simulation quality using the learned Perceptual Image Patch Similarity (LPIPS), Structural Similarity Index Measure (SSIM), and Peak Signal-to-Noise Ratio (PSNR) on a testing dataset. We evaluate task performance using environment-specific

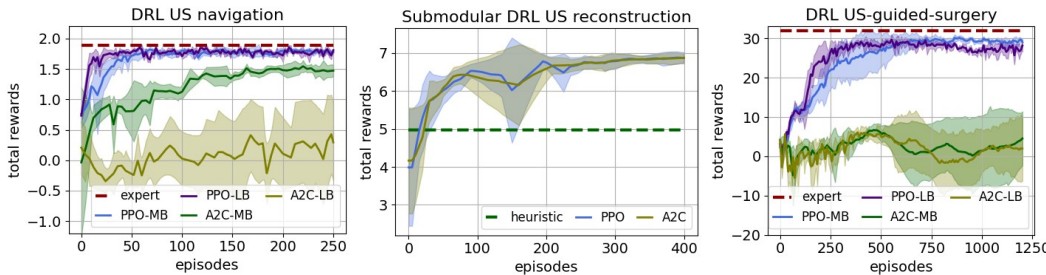

Figure 5: **Learning curves of reinforcement learning agents for all tasks.** The shaded region represents the 1-sigma confidence interval across training runs with five different random seeds. Our modeling allows stable training of PPO agents, which can achieve close performance to expert policies and better performance than A2C agents for *navigation* and *surgery* tasks. For *reconstruction*, both submodular PPO and A2C agents surpass the heuristic trajectory during the learning process.

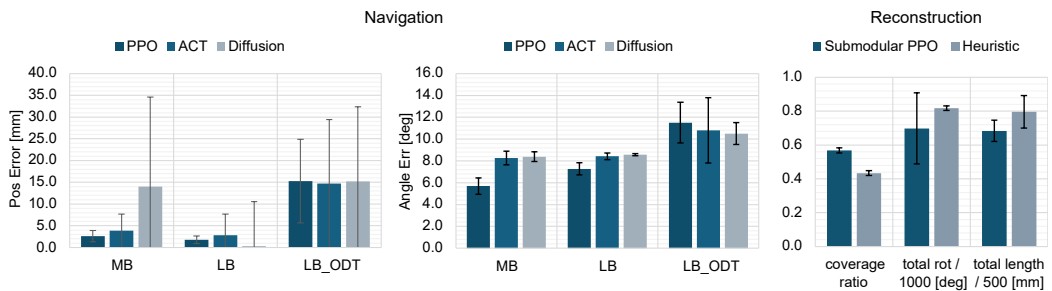

Figure 6: **Performance for *navigation* and *reconstruction*.** Results are averaged over 100 trials, and error bars denote the standard deviation. The gaps between LB_ODT and LB are not significant, which demonstrates the potential of sim-to-real transfer over the ultrasound imaging domain. For reconstruction, the submodular PPO policy surpasses the performance of the heuristic policy.

metrics: **Navigation:** Final 2D *position error* (projected onto the frontal plane of the patient) and *rotation error* (around the frontal axis); **Reconstruction:** *Coverage ratio* (reconstructed vs. total upper surface points), *total rotation angle* (pitch and yaw), and *trajectory length*; **Surgery:** *Insertion error* (position error along the $z$-axis of $\{G\}$), *side error* (position error perpendicular to the $z$-axis of $\{G\}$), *rotation error* (angle between final drill direction and $z$-axis of $\{G\}$) and *safe ratio* (the proportion of trajectory states within the safe region).

**Experiment setup**. We train both PPO and A2C agents for all environments. We also train modified SafeRPlan agents (PPO + safety filter) for the *surgery* task. For the *navigation* and *surgery* tasks, we evaluate the PPO agents on the same type of simulation used during training, corresponding to the 'MB' and 'LB' groups in Fig. 6 and In-Domain Test (IDT) columns in Tab. 1. To evaluate generalization to varying ultrasound noise conditions, we first train five ultrasound simulation networks with the same data set and different random seeds. We then train agents with 4 of these networks and test them with the 5th network, denoted as Out-of-Domain Test (ODT) columns of 'LB' rows in Tab. 1 and 'LB_ODT' in Fig. 6. For the *surgery* task, we also tested generalization across patients by training PPO and PPO + safety filter on 5 patients and evaluating on a held-out sixth patient, corresponding to the ODT columns of the 'MB' group in Tab. 1. For the reconstruction task, PPO policies were directly evaluated on the same patient and noise distribution.

### 5.1 Environment validation

**Q1: How realistic and efficient is the ultrasound simulation?** Fig. 4 presents a qualitative evaluation of the ultrasound images generated by our pix2pix network. Fig. 4 (left) shows that our network can simulate realistic images during testing. Fig. 4 (right) provides examples of both model-based (MB) and learning-based (LB) ultrasound simulations. Our LB approach maintains high visual quality despite the domain gap between the input CT slices and the training CT data. For

quantitative metrics, we achieve LPIPS loss of 0.2415, SSIM score of 0.3940, and PSNR score of 15.96, which is close to the performance of similar works reported in [9]. On an RTX 3090 Ti, the parallel rendering of ultrasound images with size $200 \times 150$ of 100 environments takes 0.0089 and 0.1107 seconds for MB and LB approaches, respectively. This enables training high-performing PPO agents within approximately 2 hours for MB simulations and 10 hours for LB simulations.

**Q2: How effective are the MDP formulations and reward design for different tasks?** Fig. 5 shows the training curves for all 3 environments with PPO and A2C. For *navigation* and *surgery* tasks, PPO can achieve close performance to the expert policy (red, based on full states) and better performance than A2C. For reconstruction, both PPO and A2C policies outperform the heuristic trajectory adopted by existing works [27]. As is also demonstrated by Fig. 6 (middle), the learned trajectory exhibits a higher coverage rate (reconstructed points divided by total points) with lower total rotation angles and path length than the heuristic policy. An example learned trajectory with a circular

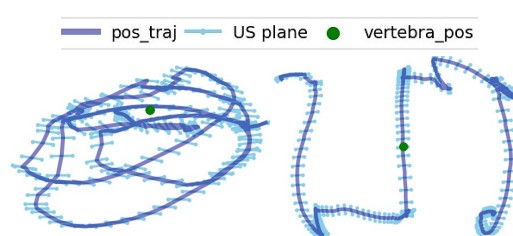

Figure 7: **Learned and heuristic trajectories for the Reconstruction task.** (left) The learned trajectory exhibits a circular pattern around the target vertebra; (right) the fixed heuristic trajectory.

shape is shown in Fig. 7 (left), which is intuitively more efficient than the heuristic trajectory (right) composed mainly of vertical and horizontal segments.

## 5.2 Comparison study

**Q3: How do RL algorithms compare with IL algorithms in different tasks?** Fig. 6 (left) and (middle) show the comparison between PPO, ACT and DP in the *navigation* task. PPO has lower position tracking variance compared to IL approaches. PPO achieves better rotation tracking accuracy. Comparison between PPO, PPO with safety filter and ACT in the *surgery* task is shown in Table. 1. In general, PPO and PPO with safety filter are more safety-aware (higher safety ratio) than ACT. They also achieves lower side position error, which can be due to the larger weight on tracking reward in the drilling region. On the contrary, ACT generally has lower safe ratios but higher tracking accuracy along the insertion direction. Fig. 8 also demonstrates that PPO trajectories are more 'conservative' by being concentrated around the goal direction and keeping a margin from the goal position.

**Q4: Can pretraining on multiple ultrasound simulation models and patients enable zero-shot generalization to unseen ultrasound noise and patients?** The generalization performance of PPO, ACT and Diffusion policy over observation domain gaps in the *navigation* task is demonstrated in Fig. 6 (left) and (middle), 'LB_ODT' groups.

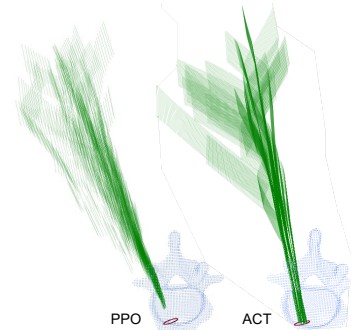

Figure 8: **Example trajectories for the *surgery* task.** The example trajectories, target vertebra and the goal position are colored green, blue, and red, respectively. PPO trajectories have less variance and stop at a certain distance from the goal, while ACT policies are less conservative and smoother.

The results show that all approaches achieve position errors of less than 16[mm] and rotation errors less than 12[deg], which is still acceptable for *navigation*. For the *surgery* task, the generalization results ('ODT' columns of 'LB' group) in general has a similar level of performance, with slightly higher *insertion error* and lower safe ratio, as is shown in Tab. 1. This shows the potential of training existing approaches with multiple ultrasound simulation networks to address the sim-to-real gap between images. Regarding testing on a new patient (ODT columns of MB in Tab. 1), both the *safe ratio* and the *side error* have a large gap to the IDT columns. This result shows that generalization across different patients is still challenging, especially with a limited diversity of patient models.

Table 1: Performance of different approaches on the *surgery* task. SF abbreviates safety filter. All values are averaged over 100 trials. The comparable performance between ODT and IDT for LB demonstrates the potential of existing approaches to generalize across the US imaging models.

| US sim | Algos | side err ↓[mm] | | insert err ↓[mm] | | rot err ↓[deg] | | safe ratio ↓[%] | |
|---|---|---|---|---|---|---|---|---|---|
| | | IDT | ODT | IDT | ODT | IDT | ODT | IDT | ODT |
| | PPO | 2.32 | **5.66** | 16.0 | 20.5 | 5.11 | 4.54 | 99.9 | 88.3 |
| MB | PPO + SF | **2.17** | 8.94 | 13.9 | 27.3 | 5.36 | **4.55** | **100.0** | **89.8** |
| | ACT | 5.38 | 18.4 | **2.92** | **14.5** | **0.62** | 5.98 | 65.9 | 65.3 |
| | PPO | 5.42 | 6.41 | 12.3 | 26.9 | 3.7 | 3.76 | 93.1 | 93.1 |
| LB | PPO + SF | **5.07** | **4.56** | 11.9 | 27.2 | 3.93 | 3.59 | **95.3** | **96.0** |
| | ACT | 5.34 | 6.26 | **1.62** | **1.72** | **1.01** | 1.81 | 86.0 | 80.3 |

# 6 Conclusion

We introduce SonoGym, a scalable simulation platform designed for complex robotic ultrasound tasks, offering fast physics-based and realistic learning-based image generation. The platform includes MDP models and expert datasets for ultrasound-guided navigation, bone reconstruction, and spinal surgery, enabling effective training of reinforcement learning and imitation learning agents. Our results highlight the potential of these approaches for sim-to-real generalization over the imaging domain gap. We also identify challenges in generalization over inter-patient variability when relying on limited patient data.

**Limitations and Future Work** Our approach has several limitations. We generated high-quality ultrasound images using a GAN-based approach; however, the GAN model may introduce artifacts and does not enforce physics-based consistency across consecutive frames. Furthermore, the patient models are static and do not capture soft-tissue deformation. The results were derived from a limited number of patient samples and have not yet been scaled to represent a broader population. In addition, our methods have not yet been validated on hardware. To address these limitations, several promising future directions can be pursued. Physics-informed losses and physics-inspired architectures could be incorporated into the generative models to improve the realism of ultrasound simulation. The dataset could be extended with more imaging parameters and anatomies, and one-step diffusion-based approaches [13, 12] could be adopted to increase the diversity of simulated ultrasound images while maintaining efficiency. Soft-tissue deformation could be incorporated using either model-based or learning-based methods like [59]. Finally, a critical next step will be validating the simulations and the RL and IL approaches on hardware.

# 7 Other Ethics Statements

Ethics approval has been granted for our CT-to-ultrasound translation dataset on cadaver by the Zurich cantonal ethics committee under BASEC Nr. 2023-01652.

# 8 Acknowledgment

This work is part of the "Learn to learn safely" project funded by a grant of the Hasler foundation (grant nr: 21039). This research was partially supported by the ETH AI Center.

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

# A  Dataset access

Below are the links to our project website, source code, simulation assets, and expert dataset.

**Project website:** `https://sonogym.github.io/`.

**Code:** `https://github.com/SonoGym/SonoGym`.

**Simulation assets:** `https://huggingface.co/datasets/yunkao/SonoGym_assets_models`.

**Expert dataset:** `https://huggingface.co/datasets/yunkao/SonoGym_lerobot_dataset`

# B  Simulation details

## B.1  Assets

We provide both the KUKA Med14 and Franka Emika Panda robot arms for ultrasound probe manipulation. To generate joint-space commands for robot control, we employ differential inverse kinematics controllers from IsaacLab. For the patient dataset, we process 3D CT images and corresponding label maps from 10 subjects in the TotalSegmentator dataset, capturing anatomical variability as illustrated in Fig. 9. Each patient's torso is converted into an STL mesh to enable physical interaction with the robot arms and patient beds in IsaacLab simulation environments. These torso models are treated as rigid bodies and are spatially aligned with their respective CT volumes and label maps. For each patient, we define the surgical target pose at the L4 vertebra using our in-house planning software, which incorporates clinical requirements for pedicle screw insertion trajectories.

## B.2  Ultrasound simulation

**Model-based approach** We follow the ray-tracing-based model introduced in [41] for ultrasound simulation. The ultrasound image $I \in \mathbb{R}^{H \times W}$ consists of a reflection component $R \in \mathbb{R}^{H \times W}$ and a backscattered component $B \in \mathbb{R}^{H \times W}$: $I = R + B$. For each 2D pixel position $u := (x, y)$, where $x \in [0, W)$ and $y \in [0, H)$, the reflection term $R$ is computed as:

$$R(u) = |E(u) \cdot \cos \Theta(u) \cdot \frac{Z(x, y + \delta y) - Z(x, y)}{Z(x, y + \delta y) + Z(x, y)}| \cdot P(u) \otimes G(u)$$

where $E(u)$ is the remaining energy at $u$, $\Theta(u)$ is the incidence angle of the ray (along the $+y$ direction) at the medium boundary surface, $Z(x, y)$ denotes the acoustic impedance of the tissue, $\delta y$ is the vertical image resolution, $P(u)$ is the point spread function (PSF), $G(u)$ is the indicator function for surface boundaries, and $\otimes$ denotes convolution. The remaining energy $E(u)$ is computed based on the attenuation coefficients $\alpha$ of the tissue at each position:

$$E(u) = E_0 \exp \left( -f \cdot \int_0^y \alpha(x, v), dv \right),$$

where $f$ is the ultrasound frequency and $E_0$ is the initial energy. The acoustic impedance $Z(u)$ is set proportional to the CT intensity, following [24]. The attenuation map $\alpha$ is determined based on the ultrasound simulation settings from Imfusion Suite [41] [2]. The PSF is modeled with a 2D Gaussian kernel with variances approximately $1\%$ of the image size. $G$ and $\Theta$ are obtained from the 2D label slice.

The backscattering term $B$ is computed as:

$$B(u) = E(u) \cdot P(u) \otimes T(u)$$

where $T(u)$ represents the scattering pattern. $T(u)$ is determined using three tissue parameters $\sigma_0$, $\mu_0$, and $\mu_1$, along with two Gaussian noise maps $N_0$ and $N_1$:

$$\tilde{T}(u) = N_0(u) \cdot \sigma_0(u) + \mu_0(u)$$
$$T(u) = \begin{cases} \tilde{T}(u), & N_1(u) \leq \mu_1(u) \\ 0, & \text{otherwise} \end{cases}$$

---

[2]https://www.imfusion.com/

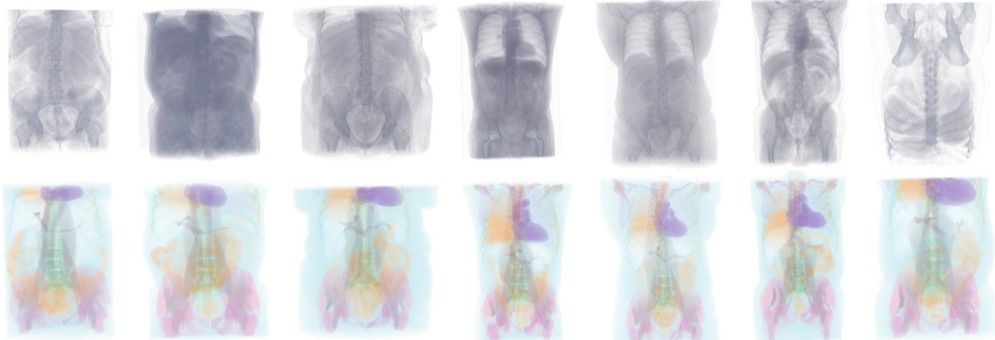

Figure 9: **Patient anatomy data from TotalSegmentator dataset.** We provide ultrasound simulation based on diverse real patient models, including CT volume and segmentation.

The noise maps $N_0$ and $N_1$ are sampled over the 3D space at initialization and remain fixed throughout the online simulation. $\sigma_0$, $\mu_0$, and $\mu_1$ of each tissue are determined based on settings in Imfusion. To capture spatial scattering variations at larger scales, we additionally incorporate multi-scale versions of $N_0$ and $N_1$, following the approach introduced in [32].

**Learning-based approach** We follow the setup described in [56] to collect a dataset of paired CT-US images from seven ex-vivo spine specimens. Optical markers are attached to the sacrum of each specimen, and additional K-wires (2.5 mm in diameter, 150 mm in length; DePuy Synthes, USA) are used to stabilize each vertebra, avoiding bone movement during data acquisition. CT scans were acquired for each specimen with an image resolution of $512 \times 512$ pixels, an in-plane pixel spacing of 0.839 mm $\times$ 0.839 mm, and a slice thickness of 0.6 mm (NAEOTOM Alpha, Siemens, Germany). For ultrasound imaging, we used the Aixplorer Ultimate system (SuperSonic Imagine, Aix-en-Provence, France) equipped with an SL18-5 linear probe (SuperSonic Imagine, Aix-en-Provence, France). An optical marker was attached to the ultrasound probe for pose tracking using an optical tracking system (FusionTrack 500, Atracsys, Switzerland). We follow the calibration pipeline in [56] to calibrate the ultrasound probe. Based on tracking data from both the spine specimen and the ultrasound probe, we register the CT and US volumes, enabling the generation of paired CT-US images.

Our pix2pix network adopts the deep U-Net architecture illustrated in Fig. 11. The model is implemented based on MONAI [5] and trained using a combination of L1 loss and GAN loss, with respective weights of 1 and 0.01. In total, we train five separate networks for 15–25 epochs on our training dataset. To improve generalization to unseen CT resolutions (such as those encountered in our simulation data), we apply data augmentation via random downsampling and upsampling.

## C  Environments details

### C.1  Task 1: Ultrasound navigation

**Environment settings** The initial 2D pose of the ultrasound probe is randomized within a $130 \times 130$ [mm$^2$] region on the frontal plane of the patient, like the region shown in Fig. 10. The orientation of the probe is initialized with a rotation between 1.5 and 3.5 [rad] from the transverse plane. We set the ultrasound image size to $200 \times 150$ pixels, with a spatial resolution of 0.5 [mm] per pixel. For the reward function, we assign a weight of $w_1 = 0.045$ to balance the position error (in [mm]) and rotation error (in [rad]).

**Agents** The network architecture used for the PPO and A2C agents is shown in Fig.12, comprising 4 convolutional layers followed by 3 fully connected layers. The agents are provided by the SKRL [45] library, which supports the

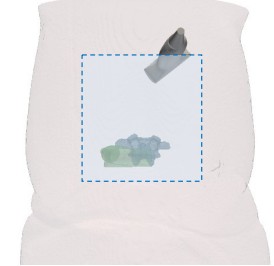

Figure 10: **Top-down view of region of manipulating the probe.**

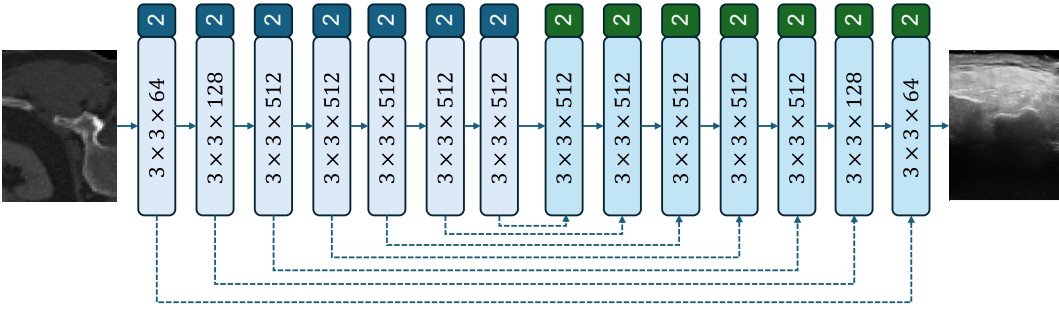

Figure 11: **U-Net architecture used for learning-based ultrasound simulation.** Dashed lines indicate residual connections within the U-Net. Convolutional (blue) and transposed convolutional (cyan) layers are annotated with (kernel size)×(kernel size)×(number of channels). The values in the blue and green blocks denote the stride used for downsampling and upsampling, respectively.

KLAdaptiveLR scheduler for the PPO agent. The hyperparameters used for training are listed in Tab.2 and Tab.3. Training is performed with 128 parallel environments, each with an episode length of 300 steps. For imitation learning, we construct expert datasets using our expert policy from three settings:

- model-based ultrasound simulation with a single patient (MB),
- learning-based simulation with a single patient (LB),
- learning-based simulation using four distinct simulation networks (LB 4 net).

The number of episodes in each dataset is 2052, 1869, and 954, respectively, as summarized in Tab. 4.

## C.2 Task 2: Bone surface reconstruction

**Environment settings** The observation volume has a shape of $40 \times 40 \times 40$ voxels with a voxel resolution of 3 [mm]. At each time step, a new 2D ultrasound image is received from the probe, and we assume that a segmentation of the bone surface is available from this image. This segmentation is simulated from the ground truth bone surface by independently applying a missing probability of 20% to each ground truth surface point. The resulting 2D segmentation map is then used to update the 3D surface reconstruction based on the current pose of the ultrasound probe. The initial 2D position of the ultrasound probe in the patient's frontal plane is randomized within a $30 \times 30$ [mm$^2$] region centered around the target vertebra position. For reward design, we use weights $w_2 = 0.01$ and $w_3 = 1$ to balance the amount of surface coverage with the total trajectory length and rotation angle.

**Agents** The network architecture of the PPO and A2C agents is illustrated in Fig.12, which contains 3 convolutional layers and 3 fully-connected layers. The hyperparameters are shown in Tab. 2 and Tab. 3. The agents are trained with 128 parallel environments with episode length 300.

Table 2: Hyperparameters of PPO agent.

| Hyperparameters | navigation | reconstruction | surgery |
|---|---|---|---|
| rollouts | 32 | 32 | 16 |
| learning_epochs | 5 | 3 | 3 |
| mini_batches | 32 | 4 | 32 |
| discount_factor | 0.99 | 0.99 | 0.99 |
| lambda | 0.95 | 0.95 | 0.95 |
| learning_rate | 0.0001 | 0.0003 | 0.0001 |
| learning_rate_scheduler: | KLAdaptiveLR | KLAdaptiveLR | KLAdaptiveLR |
| kl_threshold: 0.008 | 0.008 | 0.008 | 0.008 |
| grad_norm_clip | 1 | 1 | 1 |
| ratio_clip | 0.2 | 0.2 | 0.2 |
| value_clip | 0.2 | 0.1 | 0.1 |
| value_loss_scale | 1 | 1 | 1 |

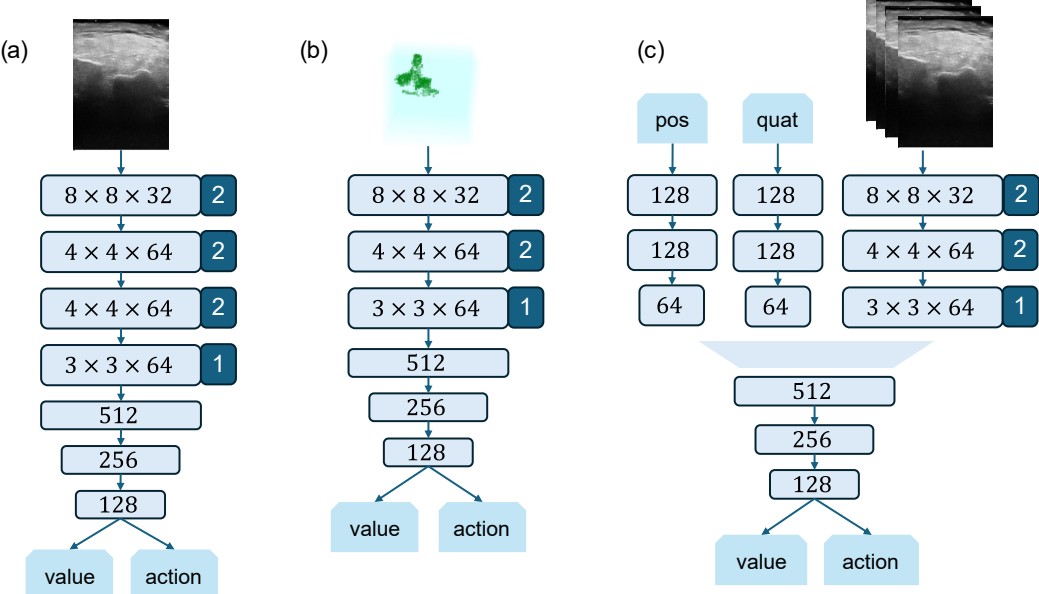

Figure 12: **Network architecture** for (a) *navigation*, (b) *reconstruction* and (c) *surgery*. The convolution layers are represented by (kernal size)×(kernal size)×(number of channels). The numbers in the blue block on the right is the numbers of strides.

Table 3: Hyperparameters of A2C agents.

| Hyperparameters | *navigation* | *reconstruction* | *surgery* |
|---|---|---|---|
| rollouts | 16 | 64 | 64 |
| learning_epochs | 5 | 3 | 1 |
| mini_batches | 32 | 32 | 4 |
| discount_factor | 0.99 | 0.99 | 0.99 |
| lambda | 0.95 | 0.95 | 0.95 |
| learning_rate | 0.0001 | 0.0001 | 0.0001 |
| grad_norm_clip | 1 | 1 | 1 |

Table 4: Number of episodes for expert datasets.

| Settings | *navigation* | *surgery* |
|---|---|---|
| MB | 2052 | 3536 |
| MB 5 patients | - | 1476 |
| LB | 1869 | 2052 |
| LB 4 net | 954 | 2692 |

### C.3 Task 3: Ultrasound-guided surgery

**Environment settings** The size of 3D ultrasound volume for the *surgery* task is $50 \times 37 \times 5$, with resolutions $2 \times 2 \times 10 [\text{mm}^3]$ along height, width, and elevation, respectively. The initial joint angles of the drill robot are randomized within $[-1.5 \pm 0.2, -0.2 \pm 0.1, 0.0 \pm 0.1, -1.3 \pm 0.1, 0.0 \pm 0.1, 1.8 \pm 0.1, 0.0 \pm 0.1]^\top [\text{rad}]$. The starting point of the trajectory on the skin is defined as $p_l = [0, 0, -50][\text{mm}]$ in the goal frame $\{G\}$. The position of the ultrasound probe above the target vertebra is set above the center of the vertebra with $30 [\text{mm}]$ translation to the ultrasound robot side. This position is further randomized within a range of 5 [mm] along all tangential axes. The reward weights are set as $w_4 = 30$, $w_5 = 5$, and $w_6 = 300$ to encourage the agent to minimize the *side error* during insertion.

**Agents** The network architecture of the PPO and A2C agents is illustrated in Fig. 12. It consists of three convolutional layers for encoding image features, three fully connected layers for encoding positional information, and three fully connected layers for encoding quaternion representations.

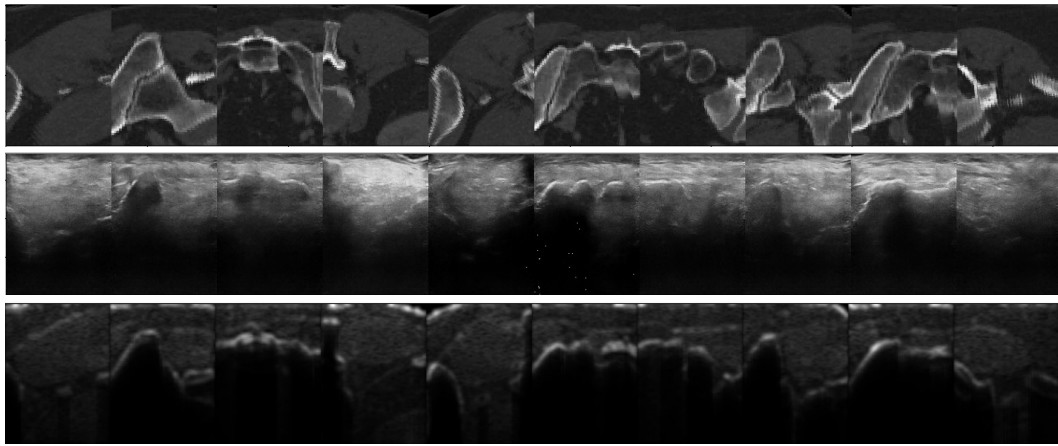

Figure 13: **Additional qualitative results of ultrasound simulation**. The top, middle, and down rows are CT slices, learning-based ultrasound simulation and model-based ultrasound simulation, respectively.

These feature streams are then concatenated and processed by an additional three fully connected layers. The training hyperparameters are summarized in Tab.2 and Tab.3. The agents are trained with 128 parallel environments with episode length of 600 steps. For the expert dataset for imitation learning, we provide datasets collected from four different settings:

- model-based ultrasound simulation from a single patient (MB),
- model-based ultrasound simulation from 5 patients (MB 5 patients),
- learning-based simulation from a single patient (LB),
- learning-based simulation with 4 simulation networks (LB 4 net).

The corresponding numbers of episodes are 3536, 1476, 2052, and 2692, respectively, as shown in Tab. 4. To increase dataset variability, we expand the range of initial robot joint angles as $[-1.6 \pm 0.3, -0.0 \pm 0.25, 0.0 \pm 0.2, -1.3 \pm 0.2, 0.0 \pm 0.2, 1.8 \pm 0.2, 0.0 \pm 0.1]^{\top}$ [rad], enabling the drill to start from configurations that may fall within unsafe regions.

## D Additional results

### D.1 Ultrasound simulation

We provide additional examples of learning-based ultrasound simulation for qualitative evaluation. Fig. 13 presents more comparisons between CT slice, model-based and learning-based ultrasound simulations. In CT slices where bone structures exhibit low contrast relative to surrounding tissues, the network occasionally struggles to generate clear bone surfaces and corresponding shadows in the ultrasound images. Although bone surfaces are clearly rendered in many cases, the bone shadows sometimes appear insufficiently dark beneath certain surfaces. We also analyze the variability across different generative networks when given the same CT slice input, as illustrated in Fig. 14. Despite all networks being trained on the identical dataset, different random seeds lead to diverse ultrasound texture patterns. This enables improved image-domain generalization of the agents by randomly sampling from multiple simulation networks during training.

### D.2 Reinforcement learning and imitation learning

**How effectively do learned and heuristic policies cover the surface?** A comparison of reconstructed surfaces between the learned and heuristic policies is shown in Fig. 15. The learned policy demonstrates greater surface coverage, particularly from the back and side views. This improvement may stem from the DRL agents learning to adjust the probe's pitch, allowing them to capture more points on surface regions where the normals are not oriented vertically.

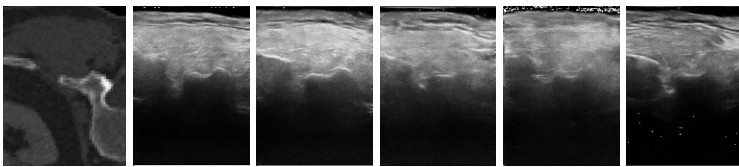

Figure 14: **Variation between ultrasound simulation models.** The input CT slice is shown left, and the other 5 ultrasound images are generated from different models with the same CT input.

Table 5: Side errors, insertion errors and rotation errors achieved by different approaches in the *surgery* task, with standard deviations included (in contrast to Tab. 1).

| US sim | Algos | side err ↓[mm] | | insert err ↓[mm] | | rot err ↓[deg] | |
| | | IDT | ODT | IDT | ODT | IDT | ODT |
|---|---|---|---|---|---|---|---|
| MB | PPO | $2.32 \pm 1.3$ | $\mathbf{5.66} \pm 18.5$ | $16.0 \pm 5.5$ | $20.5 \pm 8.0$ | $5.11 \pm 1.6$ | $4.54 \pm 4.4$ |
| | PPO + SF | $\mathbf{2.17} \pm 1.2$ | $8.94 \pm 6.1$ | $13.9 \pm 3.3$ | $27.3 \pm 5.6$ | $5.36 \pm 2.2$ | $\mathbf{4.55} \pm 2.5$ |
| | ACT | $5.38 \pm 2.7$ | $18.4 \pm 9.8$ | $\mathbf{2.92} \pm 1.1$ | $\mathbf{14.5} \pm 4.7$ | $\mathbf{0.62} \pm 0.8$ | $5.98 \pm 2.7$ |
| LB | PPO | $5.42 \pm 4.9$ | $6.41 \pm 2.6$ | $12.3 \pm 3.2$ | $26.9 \pm 4.7$ | $3.7 \pm 1.6$ | $3.76 \pm 2.5$ |
| | PPO + SF | $\mathbf{5.07} \pm 5.5$ | $\mathbf{4.56} \pm 7.0$ | $11.9 \pm 7.5$ | $27.2 \pm 4.1$ | $3.93 \pm 2.5$ | $3.59 \pm 1.8$ |
| | ACT | $5.34 \pm 3.0$ | $6.26 \pm 2.6$ | $\mathbf{1.62} \pm 0.9$ | $\mathbf{1.72} \pm 1.4$ | $\mathbf{1.01} \pm 1.0$ | $\mathbf{1.81} \pm 1.1$ |

**How well do agents generalize to a new patient with learning-based ultrasound simulation?** As shown in Tab.6, the performance on a previously unseen patient remains significantly lower compared to the results in Tab.5. This highlights the challenge of achieving generalization across anatomical variations when training with a limited number of patient examples.

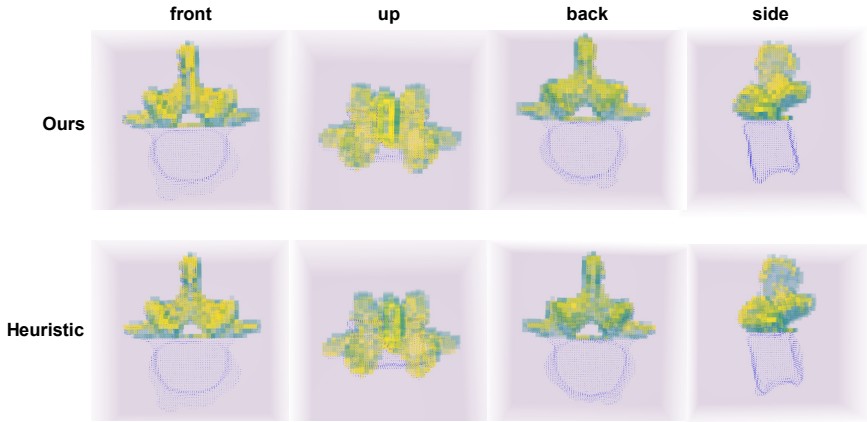

Figure 15: **Reconstructed surfaces.** The reconstructed points and uncovered points on the bone surface are colored yellow and green, respectively. The DRL policy has higher coverage from the back and side views.

**What about other agents?** We also trained Soft Actor-Critic (SAC) agents for both the *navigation* and *surgery* tasks, but were unable to obtain high-performing policies. We adopted the same network architecture as in our PPO/A2C experiments, but without sharing the encoder between the policy and value networks. Our hyperparameter search covered the following ranges: actor learning rate (0.00001, 0.0001, 0.001), critic learning rate (0.00001, 0.0001, 0.001), gradient steps (1, 8, 32), and batch size (64, 256). We set the replay buffer size to 64,000, taking into account the high memory usage of image observations.

For the *surgery* task, we additionally trained PPO-Lagrangian agents using the following hyperparameter search ranges: learning rates 0.0001 with KLAdaptiveLR scheduler, number of rollouts (16, 32, 64), number of mini-batches (4, 32, 128), number of learning epochs (1, 3, 5), and value loss scale (1.0, 3.0, 10.0). However, none of these configurations resulted in consistently successful policies.

Table 6: Performance of different approaches on a new patient for the *surgery* task with learning-based ultrasound simulation

| Algos | side err [mm] | insert err [mm] | rot err [deg] | safe ratio [%] |
|---|---|---|---|---|
| PPO | 27.3 ±36.4 | 12.1± 8.77 | 8.28± 7.33 | 52.86 |
| PPO + SF | 17.6 ±54.2 | 11.2 ±7.48 | 8.02 ±7.12 | 60.08 |
| ACT | 20.7± 15.6 | 15.4 ±3.92 | 7.64 ±2.85 | 72.06 |

While Decision Transformer (DP) achieved promising results for the *navigation* task, it failed to perform well on the *surgery* task using the default settings. We experimented with varying the number of historical observation steps (1, 2), action steps (4, 8), planning horizons (8, 16), and learning rates (0.00001, 0.0001), but these adjustments did not lead to significant performance improvements.

