# OpenReview forum: "SonoGym: High Performance Simulation for Challenging Surgical Tasks with Robotic Ultrasound"
_NeurIPS.cc/2025/Datasets_and_Benchmarks_Track — NeurIPS 2025 Datasets and Benchmarks Track poster_

### Official Review · Reviewer_xLUZ · 2025-06-13

**Rating:** 4
**Confidence:** 2

**Summary:**

This paper represents a Ultrasound simulation platform for three tasks: ultrasound navigation, bone surface reconstruction, ultrasound-guided spinal surgery. The ultrasound simulation is implemented by both model-based and learning based methods according to the CT information. Typical reinforcement learning methods and imitation learning methods are achieved for three tasks and show good performance.

**Dataset Code Accessibility:**

Yes

**Ethical Considerations:**

No, there are no or only very minor ethics concerns

**Final Justification:**

The authors solve my concerns, and I keep my original score.

**Limitations Weaknesses:**

1. How many data do you use to train the learning-based method (GAN model)?
2. Although the paper mentioned that the reason for using GAN-based method is fast, diffusion-based method is supposed to have better quality, and some accelerating techniques could be used. For example, this paper "Dynamic Diffusion Transformer" proposed a method to accelerate the diffusion progress by 50%.

**Strengths Contributions:**

1. Reinforcement learning methods and imitation learning methods are designed for three typical ultrasound-related tasks, including ultrasound navigation, bone surface reconstruction, and ultrasound-guided spinal surgery.
2. The experiments show that the reinforcement learning methods achieve good performance regarding the expert model. Some generalization ability is shown in the navigation task.
3. The simulation speed is relatively fast.

---

> ### Author Rebuttal · Authors · 2025-07-31
>
> **Response to Summary and Strength Contributions**
>
> We appreciate the reviewer’s positive feedback and recognition of our contributions.
>
> **Response to Limitations Weaknesses**
>
> >```
> > How many data do you use to train the learning-based method (GAN model)?
>
>
> We thank the reviewer for the question. Our training dataset contains 111429 pairs of CT and ultrasound images.  We will include this detail in the later version of the manuscript.
>
> >```
> >Although the paper mentioned that the reason for using GAN-based method is fast, diffusion-based method is supposed to have better quality, and some accelerating techniques could be used. For example, this paper "Dynamic Diffusion Transformer" proposed a method to accelerate the diffusion progress by 50%.
>
>
> We thank the reviewer for the thoughtful comments. We fully agree that diffusion models offer advantages in terms of image quality and diversity for ultrasound simulation, and we have observed promising results in a preliminary study. However, one key limitation is their computational cost: the multi-step sampling process inherent to standard diffusion models makes them difficult to use directly in environments of deep reinforcement learning (RL), where high-throughput simulation is essential.
>
> In fact, even with GAN-based image generation, our parallel simulation is already slower compared to model-based approaches. This highlights the need for efficient alternatives. We are particularly interested in exploring fast or one-step diffusion models, which may offer a good balance between realism and computational efficiency. Promising methods can include _Dynamic Diffusion Transformers_ [1], _Mean Flow Matching_ [2], and _Shortcut Models_ [3].
>
> That said, the application of these techniques to ultrasound image generation remains largely unexplored. Investigating their potential in this context was out of the scope for this paper, but it will be an interesting research direction built on top of our platform.
>
> **References**
>
> [1] Zhao, Wangbo, et al. "Dynamic diffusion transformer." arXiv preprint arXiv:2410.03456 (2024).
>
> [2] Geng, Zhengyang, et al. "Mean flows for one-step generative modeling." arXiv preprint arXiv:2505.13447 (2025).
>
> [3] Frans, Kevin, et al. "One step diffusion via shortcut models." arXiv preprint arXiv:2410.12557 (2024).

---

### Official Review · Reviewer_QwPn · 2025-06-30

**Rating:** 4
**Confidence:** 1

**Summary:**

This paper proposes SonoGym, a scalable simulation platform for robotic ultrasound, which supports real-time parallel ultrasound image simulation based on both physics-based models and Generative Adversarial Networks (GANs). The platform enables the training of reinforcement learning (RL) and imitation learning (IL) agents (e.g., PPO, ACT, and Diffusion Policy) for three key orthopedic surgical tasks: navigation, anatomy reconstruction, and surgical execution. The authors further incorporate advanced learning techniques such as Safe RL and submodular MDPs, and demonstrate that the trained policies generalize well under various ultrasound noise conditions and across different patient anatomies.

**Dataset Code Accessibility:**

Yes

**Dataset Code Comments:**

The authors have made both the dataset and source code publicly available to support reproducibility and further research. The dataset can be accessed at: https://huggingface.co/datasets/yunkao/SonoGym_lerobot_dataset, and the code repository is available at: https://github.com/SonoGym/SonoGym.

**Ethical Considerations:**

No, there are no or only very minor ethics concerns

**Final Justification:**

I maintain my current positive rating.

**Limitations Weaknesses:**

1. Although the simulated ultrasound is realistic, the paper does not provide validation results on real robotic deployments or human-patient interaction. It would be beneficial to include qualitative comparisons to expert human trajectories or real-world behaviors.

2. The reward function design—particularly for the surgical task—is complex and involves manually tuned weights. The paper would benefit from a discussion on reward robustness or automatic tuning strategies to improve generalization and reusability.

3. It is unclear whether informed consent was obtained for using real patient data. This point should be clarified for ethical transparency.

**Strengths Contributions:**

1. This work targets robotic ultrasound in orthopedic surgery, a clinically relevant area that currently lacks strong simulation platform support. The paper is highly relevant to real-world surgical applications.

2. The platform supports both physics-based and learning-based ultrasound simulation with high parallel scalability.

---

> ### Author Rebuttal · Authors · 2025-07-31
>
> **Response to Summary and Strength Contributions**
>
> We thank the reviewer for the thorough reading and insightful comments.
>
>
> **Response to Limitations Weaknesses**
>
> > ```
> >1. Although the simulated ultrasound is realistic, the paper does not provide validation results on real robotic deployments or human-patient interaction. It would be beneficial to include qualitative comparisons to expert human trajectories or real-world behaviors.
>
>
> We appreciate the reviewer’s thoughtful comments. We agree that validation through real robotic deployment would significantly strengthen the contribution of our work. However, we note that the complex physical interaction between the robot and the human body introduces additional challenges for achieving reliable sim-to-real transfer. As has been mentioned in the response to Reviewer 2, several key research questions need to be addressed:
> - How can we develop a robust low-level controller that ensures safe and stable contact between the probe and the skin while maintaining a constant force?
> - How can we enable effective generalization from synthetic to real ultrasound images, given the potential differences in imaging parameters and acquisition conditions?
> - Are raw ultrasound images the most suitable observation space for policy learning, or would more abstract representations (like segmentations or CT-based projections) provide better performance and robustness?
> - Can a policy be trained to remain robust under real-time anatomical deformations, such as those caused by respiratory motion?
>
> These questions are central to our ongoing and future research, and we aim to address them as part of our long-term roadmap.
>
> We also acknowledge the reviewer’s suggestion regarding trajectory comparison. Comparing learned trajectories with expert human demonstrations is indeed a meaningful form of evaluation. For the surface reconstruction task, we compared our method against a heuristic trajectory commonly used in existing robotic reconstruction work. We agree that incorporating expert human trajectories would provide valuable additional insight. Since such demonstrations require data collection on physical phantoms, we plan to include this in our future sim-to-real validation efforts. We will also include this point in the later version of our manuscript as a future plan.
>
> For our navigation and ultrasound-guided surgery tasks, we can actually define ground-truth optimal trajectories:
> For navigation, the optimal trajectory is defined as a direct path to the target location.
> For surgery, the trajectory aligns with the preoperative plan.
> Nonetheless, once a real robotic system is in place, it will be interesting to compare the performance of our policies against expert surgeons in practice. We leave this as part of our future efforts.
>
>
> > ```
> >2. The reward function design—particularly for the surgical task—is complex and involves manually tuned weights. The paper would benefit from a discussion on reward robustness or automatic tuning strategies to improve generalization and reusability.
>
>
> We agree that our reward design requires careful engineering effort, particularly given the complexity of the surgical task. In our case, the agent must navigate through a narrow safe corridor surrounded by unsafe regions. This makes the reward design challenging to guide the policy to achieve the goal safely. To achieve this, we carefully tuned the reward components to ensure that the trained agents execute the desired motion.
>
> Additionally, the differing scales of translation and rotation errors required appropriate weighting to balance their influence on the final performance. Due to the space limit, the detailed choices of reward weights have been provided in the appendix of the original manuscript.
>
> We also agree with the reviewer that automating the exploration of reward weight configurations would be beneficial. We plan to incorporate such functionality into our simulation platform in future releases. Besides, using LLM-based reward design like [1] might also be an interesting direction.
>
>
> > ```
> >3. It is unclear whether informed consent was obtained for using real patient data. This point should be clarified for ethical transparency.
>
>
> We thank the reviewer for the question. The real patient data used in our study is an open-source dataset, ‘TotalSegmentator dataset’, published by University Hospital Basel. Regarding the CT-to-ultrasound translation dataset on cadaver, ethics approval has been granted for our study by the Zurich cantonal ethics committee under BASEC Nr. 2023-01652.
>
> **References**
>
> [1] Ma, Yecheng Jason, et al. "Eureka: Human-level reward design via coding large language models." arXiv preprint arXiv:2310.12931 (2023).

---

> > ### Comment · Reviewer_QwPn · 2025-08-02
> > **After Rebuttal**
> >
> > Thank you for the authors’ response, which has largely addressed my concerns. My overall evaluation remains positive.

---

### Official Review · Reviewer_UTaG · 2025-07-02

**Rating:** 4
**Confidence:** 2

**Summary:**

This paper introduces SonoGym, a scalable simulation platform designed to facilitate research into robotic ultrasound-guided surgical tasks, with a particular focus on orthopaedic applications such as spinal navigation, bone surface reconstruction and ultrasound-guided drilling surgery.
The platform integrates both physics- and GAN-based ultrasound (US) simulations derived from computed tomography (CT) scans of patients, supports parallelised training across hundreds of environments and provides Markov decision process (MDP) formulations for three clinically relevant tasks: navigation, reconstruction and surgery. Benchmark experiments using Reinforcement Learning (PPO, A2C and SafeRL variants) and Imitation Learning (Action Chunking Transformers and Diffusion Policies) are conducted on these tasks to demonstrate the value of SonoGym.

**Dataset Code Accessibility:**

Yes

**Dataset Code Comments:**

The dataset and code are publicly available on Hugging Face and GitHub, respectively.

**Ethical Considerations:**

No, there are no or only very minor ethics concerns

**Final Justification:**

The authors' responses to the limitations and weaknesses are satisfactory.

**Limitations Weaknesses:**

Limited patient diversity and generalisation:

- Experiments use only data from five to six patients. Results from cross-patient testing show significant performance degradation, particularly in surgery. This calls into question the robustness of the technology for clinical deployment.

GAN simulation constraints:

- GANs often undergo unstable training, which may compromise the representativeness of the GAN-generated data. Additionally, GANs lack physics-based regularisation. Artifacts may mislead policy learning (e.g. false bone edges). There is no validation of how well the learned policies transfer from GAN-simulated to real ultrasound (US) data.

Preliminary real-world validation:

- No real robot/surgery validation. Policies are trained and tested purely in simulation.  There has been no demonstration on physical robots or phantoms, even for core navigation and reconstruction tasks. This limits confidence in the real-world applicability of the learned behaviours.

**Strengths Contributions:**

Significance and impact:

- The SonoGym fills a critical gap in surgical robotics simulation by providing the first unified platform for ultrasound-guided reconstruction and surgery (beyond navigation), thereby enabling research into high-stakes autonomous surgical tasks.

- Clinically grounded tasks: Well-defined MDPs for navigation (see Fig. 3, left), reconstruction (submodular rewards) and safety-constrained surgery (see Fig. 3, right) align with orthopaedic requirements. Integration with the IsaacLab robotics stack enhances practicality.

- Comprehensive benchmarking: Performance comparison of RL (PPO and A2C) and IL (ACT and DP) agents across all tasks, with additional analysis of generalisation capabilities.

Novelty and technical quality:

- Safety-aware surgery: Modified SafeRL (SafeRPlan) ensures  a high safety ratio (>95%) in drilling tasks (see Table 1), which is a crucial advancement for clinical translation.

- GAN for RL training: Pioneering use of GAN-simulated ultrasound (US) images (trained on specific ex vivo spine data) to train RL/IL agents in a surgical context. This shows promising sim-to-real generalisation potential for the imaging modality itself.

---

> ### Author Rebuttal · Authors · 2025-07-31
>
> **Response to Summary and Strength Contributions**
>
> We appreciate the reviewer for the thorough reading and highlighting of our contributions.
>
> **Response to Limitations Weaknesses**
> > ```
> > Limited patient diversity and generalisation: Experiments use only data from five to six patients. Results from cross-patient testing show significant performance degradation, particularly in surgery. This calls into question the robustness of the technology for clinical deployment.
>
>
> We thank the reviewer for the thoughtful comments. We fully agree that increasing the diversity of patient models during training could also enhance generalization performance. Through our results, we aimed to highlight that generalizing across patient variability using only a limited number of patient models remains a significant challenge. This presents opportunities for further research into more advanced algorithmic solutions. One goal of this work is to initiate community interest in these tasks, and all advanced features will be added with time through us as well as through the community.
>
> As a core focus of our future work and mentioned in the ‘Conclusion’ section, we will improve our platform to enable training with a broader range of patient models. For instance, we can gradually incorporate more patient models from the TotalSegmentator dataset or the Ultrabones100k [1] dataset to expand the anatomical region and surgical tasks.
>
> >```
> > GAN simulation constraints: GANs often undergo unstable training, which may compromise the representativeness of the GAN-generated data. Additionally, GANs lack physics-based regularisation. Artifacts may mislead policy learning (e.g. false bone edges). There is no validation of how well the learned policies transfer from GAN-simulated to real ultrasound (US) data.
>
> We agree that GAN-based approaches have several limitations, including unstable training, reduced physical consistency, and mode collapse. To address these challenges, we consider fast diffusion-based models and physics-informed methods as promising alternatives for improving ultrasound image generation. However, their application in this domain remains relatively underexplored. Besides, the main focus of this paper is to provide a simulation platform to allow researchers to test different approaches for the robotic ultrasound tasks. Therefore, we consider investigating novel approaches for ultrasound imaging generation out of scope for this work. But we plan to investigate these directions as part of our future work to improve the ultrasound simulation quality and noise diversity, as noted in the ‘Conclusion’ section.
>
> We also agree that validation with real ultrasound data would significantly strengthen the contribution. In the original version of the paper, we include comparisons between generated and real ultrasound images. Beyond that, we are actively working on extending our system and conducting validation experiments on a real robotic platform as part of our ongoing efforts for future work.
>
>
>
> > ```
> > Preliminary real-world validation: No real robot/surgery validation. Policies are trained and tested purely in simulation. There has been no demonstration on physical robots or phantoms, even for core navigation and reconstruction tasks. This limits confidence in the real-world applicability of the learned behaviours.
>
>
> We thank the reviewer for pointing out this important limitation. Indeed, validation on physical robots and phantoms would provide a stronger demonstration of the applicability of the learned policy. We have discussed this limitation in the ‘Conclusion’ section. In this work, however, our primary contribution is the development and open-sourcing of a simulation platform designed to support and accelerate research in this direction.
>
> Achieving sim-to-real transfer for the proposed tasks is currently challenging because it requires investigating additional research questions. For example:
> - How to develop a robust and adaptive low-level controller that maintains stable and safe contact between the probe and the skin with a constant force?
> - How can we enable generalization from synthetic to real ultrasound images, which may differ in imaging parameters and acquisition conditions?
> - Are raw ultrasound images the most effective observation space for the policy, or would more abstract representations (e.g., segmentations, CT projections) offer better generalization performance?
> - Is it possible to train a policy that remains robust to real-time deformations of the patient’s body, such as those caused by breathing motion?
>
> These are interesting topics of our ongoing and future research. We also hope that the open-source nature of our platform will encourage the broader research community to explore these questions further. Since there has been success of sim-to-real transfer for manipulation tasks [2, 3, 4], we consider it feasible to achieve similar results with our platform in the near future.
>
> **References**
>
> [1] Wu, Luohong, et al. "UltraBones100k: A reliable automated labeling method and large-scale dataset for ultrasound-based bone surface extraction." Computers in Biology and Medicine 194 (2025): 110435.
>
> [2] Narang, Yashraj, et al. "Factory: Fast contact for robotic assembly." arXiv preprint arXiv:2205.03532 (2022).
>
> [3] Tang, Bingjie, et al. "Automate: Specialist and generalist assembly policies over diverse geometries." arXiv preprint arXiv:2407.08028 1.2 (2024).
>
> [4] Handa, Ankur, et al. "Dextreme: Transfer of agile in-hand manipulation from simulation to reality." 2023 IEEE International Conference on Robotics and Automation (ICRA). IEEE, 2023.

---

### Official Review · Reviewer_nCZp · 2025-07-10

**Rating:** 6
**Confidence:** 5

**Summary:**

This submission presents SonoGym, a scalable and high-performance simulation platform specifically designed for challenging robotic ultrasound tasks in surgical settings. The primary contributions include the development of realistic and efficient ultrasound simulation environments, which utilize both model-based and GAN-based methods to generate ultrasound images from real patient CT datasets. SonoGym enables the parallel simulation of tens to hundreds of environments and supports multiple surgical tasks, including ultrasound probe navigation, bone surface reconstruction, and ultrasound-guided spinal surgery. The platform provides standard Markov Decision Process (MDP) formulations for these tasks, allowing the benchmarking and training of state-of-the-art reinforcement learning (RL) and imitation learning (IL) algorithms, including vision transformers and diffusion policies. Comprehensive experiments demonstrate the ability of RL and IL methods to achieve robust policy learning across various scenarios, as well as the potential for sim-to-real transfer. All code, datasets, and videos are made publicly available to promote further research in autonomous robotic surgery.

**Additional Feedback:**

The paper presents a well-executed and timely contribution to the field of medical robot learning. The design of SonoGym, with its support for multiple tasks and algorithms, will likely foster new research directions in autonomous surgery and ultrasound-based navigation. The public release of code and data is highly commendable and sets a strong standard for transparency and reproducibility.

---


**Suggestions for improvement:**

* Consider expanding the diversity of patient anatomies in future versions, as this will improve the generalizability of both the simulation platform and the agents trained with it.
* Adding support for soft-tissue deformation would make the simulation environment even more realistic and valuable for a wider range of clinical use-cases.
* Including at least a small-scale real-world validation, even if preliminary, would greatly strengthen the practical impact of the work and the case for sim-to-real transfer.
* In future updates, it may be helpful to provide more detailed documentation or tutorials, including example workflows for common tasks (e.g., agent training, evaluation, or adding new anatomical models).


**Questions for the authors:**

* Are there plans to extend the platform to other anatomical regions or modalities beyond the spine and ultrasound?
* Do you intend to host a leaderboard or regular benchmark challenges to encourage community engagement and continued improvement?

Overall, this is a strong and well-organized submission that will be of significant value to the research community. Thank you for your efforts in making the resources open and accessible.

**Dataset Code Accessibility:**

Yes

**Dataset Code Comments:**

The submission provides open access to all datasets, code, and documentation via the project website (https://sonogym.github.io/), as clearly stated in both the abstract and the main text. The data is available in its final, usable form, and the codebase includes comprehensive documentation and example scripts to facilitate reproducibility. Detailed instructions for installation, usage, and experimental replication are provided, supporting easy adoption and benchmarking by the community. The inclusion of both model-based and GAN-based simulation code, as well as expert demonstration datasets, further ensures that researchers can fully reproduce the reported experiments and extend the benchmark suite.

**Ethical Comments:**

The submission does not involve improper research with human subjects, nor does it introduce significant concerns regarding data privacy, consent, or copyright, as it leverages publicly available datasets (such as TotalSegmentator) and ex-vivo data with appropriate citation. The dataset and code are released for research purposes and do not pose obvious risks related to safety, security, or discrimination. The data representativeness is discussed, with limitations acknowledged regarding patient diversity, but no evidence suggests intentional or negligent exclusion or bias. There are no issues related to environmental impact or human rights. Overall, the work adheres to NeurIPS ethical guidelines, and no significant ethical concerns remain.

**Ethical Considerations:**

No, there are no or only very minor ethics concerns

**Final Justification:**

Through the rebuttal phase, I think the authors address my major concerns.

**Limitations Weaknesses:**

Despite its notable contributions, the work also presents several limitations that should be acknowledged and addressed in future iterations. First, as discussed in the Conclusion and Limitations sections (pages 9 and 13), the patient models used in SonoGym are static and do not incorporate soft tissue deformation. This restricts the simulation’s realism, especially for procedures where tissue compliance plays a critical role. Extending the framework to support deformable tissue models, possibly by integrating existing soft-tissue simulation toolkits (e.g., SOFA), would increase clinical fidelity and broaden applicability.

Second, the learning-based ultrasound simulation (LB) relies on a GAN model trained on a relatively small, in-house CT-to-ultrasound paired dataset from just seven ex-vivo specimens (Section 4, page 4). This limited diversity may reduce the robustness and generalizability of learned agents, as also evidenced by the decreased performance when generalizing to unseen patients (see Table 1, ODT columns, page 9). Expanding the dataset to include more diverse patient anatomies and clinical conditions would be a concrete way to improve both simulation realism and agent robustness.

Third, although the paper demonstrates successful sim-to-real transfer for varying ultrasound imaging noise (domain shift within the simulation), it also highlights persistent challenges in generalizing across different patient anatomies (Section 5.2 and Table 1, page 9). Future work could incorporate advanced domain randomization techniques or real patient data augmentation to address this gap.

Fourth, the current GAN-based simulation method does not explicitly model physical constraints, which could limit the interpretability and reliability of generated ultrasound images in edge cases (Limitations, page 9). Combining GANs with physics-informed losses or hybrid approaches might further improve realism.

Lastly, all evaluations are currently limited to the simulation environment. There is no validation with real-world robotic systems or in clinical settings. Although this is common in early-stage simulation work, providing even preliminary results or pilot experiments with hardware could greatly strengthen the claims regarding sim-to-real applicability.


---


**Actionable Suggestions:**

* Integrate deformable tissue models for higher anatomical realism.
* Expand the dataset to cover more patient diversity and clinical variability.
* Investigate advanced domain randomization or data augmentation to improve generalization.
* Explore hybrid simulation approaches combining physics-based and learning-based models.
* Plan or report preliminary validation with real robotic hardware, if feasible.

By addressing these limitations, the SonoGym framework could become an even more powerful and reliable resource for the research community.

**Strengths Contributions:**

The paper’s key strengths lie in its comprehensive simulation framework and clear focus on enabling high-impact research for robotic ultrasound in surgery. SonoGym is distinguished by its support for multiple clinically relevant tasks, namely navigation, bone surface reconstruction, and ultrasound-guided surgery, using both model-based and advanced GAN-based simulation methods. This dual approach enables both realism and computational efficiency, as demonstrated by the quantitative results on simulation quality (e.g., SSIM, LPIPS, PSNR) and the ability to render hundreds of environments in parallel (Section 5.1, Figure 4).

A significant contribution is the integration of real patient CT data and anatomical segmentations from the TotalSegmentator dataset, resulting in simulation environments that closely mirror actual surgical contexts. The public release of the code, datasets, and pre-collected expert demonstrations further increases the potential impact, lowering the barrier for adoption and benchmarking across the community.

The benchmark suite is novel and well-justified: it covers a broad spectrum of state-of-the-art learning algorithms, including PPO, submodular RL, vision transformers, and diffusion policies. The extensive experimental analysis demonstrates both the strengths and current limitations of RL/IL in this domain, especially regarding generalization to unseen imaging conditions and patient anatomies (Table 1 and Figure 6). Compared to prior work, the manuscript clearly articulates how SonoGym expands on previous simulation platforms (such as Isaac Healthcare, LapGym, and SurRol), by uniquely focusing on robotic ultrasound, clinical realism, and open-access resources (Related Works, Section 2).

The presentation quality is high: the paper is well-written, logically organized, and supported by clear diagrams and informative tables (e.g., the visual task overviews in Figure 3 and the quantitative results in Table 1). Captions and figure references are consistently clear and helpful, and the provided website ensures reproducibility and further exploration.

In summary, SonoGym stands out as a significant contribution to the fields of medical robotics and robot learning, providing the community with a realistic, extensible, and open simulation environment for developing and benchmarking autonomous surgical agents.

---

> ### Author Rebuttal · Authors · 2025-07-31
>
> **Response to Summary and Strength Contributions**
>
>
> We thank the reviewer for the thorough reading and insightful comments. We are glad that the reviewer found our platform valuable for future research on robot learning and medical robotics.
>
> **Response to limitation weaknesses**
> >```
> >Despite its notable contributions, the work also presents several limitations that should be acknowledged and addressed in future iterations. First, as discussed in the Conclusion and Limitations sections (pages 9 and 13), the patient models used in SonoGym are static and do not incorporate soft tissue deformation. This restricts the simulation’s realism, especially for procedures where tissue compliance plays a critical role. Extending the framework to support deformable tissue models, possibly by integrating existing soft-tissue simulation toolkits (e.g., SOFA), would increase clinical fidelity and broaden applicability.
>
>
> We thank the reviewer for highlighting this important point. Indeed, soft-tissue deformation is not currently supported in our framework, and incorporating it could further enhance the applicability of our platform. We have discussed this in our ‘Conclusion’ section of the original manuscript. A key challenge in this regard is balancing simulation accuracy with computational efficiency. As a promising future direction, we are particularly interested in integrating learning-based simulation methods (like approximating finite element analysis with neural network [1]) to address this trade-off.
>
> >```
> >Second, the learning-based ultrasound simulation (LB) relies on a GAN model trained on a relatively small, in-house CT-to-ultrasound paired dataset from just seven ex-vivo specimens (Section 4, page 4). This limited diversity may reduce the robustness and generalizability of learned agents, as also evidenced by the decreased performance when generalizing to unseen patients (see Table 1, ODT columns, page 9). Expanding the dataset to include more diverse patient anatomies and clinical conditions would be a concrete way to improve both simulation realism and agent robustness.
>
>
> We thank the reviewer for the insightful comments.  While, to the best of our knowledge, there is no open-source large-scale CT-to-ultrasound dataset for the spine, we rely on our in-house dataset (an extended version of [2]) as the most suitable resource currently available. However, we acknowledge that the anatomical variability within this dataset remains limited for training our CT-to-ultrasound generative network. We will include the discussion on this limitation in our later version. As part of our ongoing efforts at Balgrist University Hospital, we are expanding our dataset to include more anatomies and ultrasound imaging parameters. More powerful models will be trained once more open-source or in-house datasets are available.
>
> >```
> >Third, although the paper demonstrates successful sim-to-real transfer for varying ultrasound imaging noise (domain shift within the simulation), it also highlights persistent challenges in generalizing across different patient anatomies (Section 5.2 and Table 1, page 9). Future work could incorporate advanced domain randomization techniques or real patient data augmentation to address this gap.
>
>
> We thank the reviewer for pointing out this important challenge. We have discussed this point in the ‘Conclusion’ section of the original manuscript. Regarding generalization across patient anatomies, we agree that incorporating advanced domain randomization and data augmentation techniques could be valuable research directions based on our platform. We are also working toward supporting more patient models within our simulation platform to further improve variability during training.
>
> >```
> >Fourth, the current GAN-based simulation method does not explicitly model physical constraints, which could limit the interpretability and reliability of generated ultrasound images in edge cases (Limitations, page 9). Combining GANs with physics-informed losses or hybrid approaches might further improve realism.
>
>
> We acknowledge that the reviewer highlights this limitation. We would like to emphasize that the main contribution of our paper is to provide a simulation platform that allows the community to test different algorithms. Therefore, we consider it out of scope for this paper to investigate physics-informed loss and physics-inspired network architectures for more diverse and realistic ultrasound imaging generation, but they will be interesting directions for further work.
>
> >```
> >Lastly, all evaluations are currently limited to the simulation environment. There is no validation with real-world robotic systems or in clinical settings. Although this is common in early-stage simulation work, providing even preliminary results or pilot experiments with hardware could greatly strengthen the claims regarding sim-to-real applicability.
>
>
> We thank the reviewer for the constructive and thoughtful comments. We are indeed actively working on hardware integration and sim-to-real transfer for the proposed tasks. A critical requirement is the development of a robust low-level controller that satisfies the assumptions made in our task formulation. Specifically, the controller should be able to maintain stable and safe contact between the probe and the skin with a constant force. We note that this component is itself non-trivial and requires additional dedicated research and engineering effort.
>
> To address the sim-to-real gap in observations, we also consider it important to introduce more diverse domain randomization into ultrasound image simulations. This may involve collecting additional datasets acquired under varying ultrasound imaging parameters. Besides, GAN-based methods are limited by mode collapse, which constrains the diversity of simulated images. As a promising alternative, one-step diffusion models could provide improved image diversity and realism, though they have not yet been explored in the context of ultrasound image synthesis and require further investigation.
>
> Currently, we consider the sim-to-real transfer to be beyond the scope of this work, as our main focus is providing the open-source benchmarking environments to facilitate research in the medical robotics field. However, advancing sim-to-real transfer remains a core objective of our ongoing and future work.
>
> **Response to actionable suggestions and suggestions for improvements**
>
>
> We appreciate the reviewer’s constructive suggestions, many of which are well aligned with our ongoing and planned efforts. Specifically, as future directions, we intend to:
> - Optimize the workflow to broaden the range of supported patient models
> - Improve the physical plausibility and variability of learning-based ultrasound simulations.
> - Expand the CT-to-ultrasound dataset to other imaging parameters and anatomical regions.
> - Improve generalization over observation domains and anatomical variation
> - Prepare preliminary studies on real robotic hardware
> - Provide more detailed documentation for the framework
> - Integrate soft-tissue simulation
>
>
> **Response to questions for the authors**
> > ```
> > 1. Are there plans to extend the platform to other anatomical regions or modalities beyond the spine and ultrasound?
>
>
> We thank the reviewer for the question. Regarding the extension to additional anatomical regions, we believe it is currently feasible to adapt our platform for foot-and-ankle surgery using the Ultrabones100k dataset [2]. We are also working on collecting datasets of the lower limbs (tibia, fibula) and the hips (femur + pelvis), which will further expand the range of applicable anatomical regions. Furthermore, we see strong potential in incorporating other imaging modalities (such as intraoperative X-rays) through the style transfer methods [3]. These directions are part of our long-term development plan, and they will be gradually integrated into the platform.
>
> > ```
> > 2. Do you intend to host a leaderboard or regular benchmark challenges to encourage community engagement and continued improvement?
>
>
> We agree that this is a valuable idea to foster greater community engagement and collaborative improvement. We will consider incorporating it as part of our long-term plan.
>
> **References**
>
> [1] Yuan, Xintian, et al. "MIXPINN: Mixed-Material Simulations by Physics-Informed Neural Network." arXiv preprint arXiv:2503.13123 (2025).
>
> [2] Wu, Luohong, et al. "UltraBones100k: A reliable automated labeling method and large-scale dataset for ultrasound-based bone surface extraction." Computers in Biology and Medicine 194 (2025): 110435.
>
> [3] Jecklin, Sascha, et al. "Domain adaptation strategies for 3D reconstruction of the lumbar spine using real fluoroscopy data." Medical Image Analysis 98 (2024): 103322.

---

> > ### Comment · Reviewer_nCZp · 2025-08-05
> > **Author Response**
> >
> > I would like to express my sincere gratitude to the authors for providing the detailed responses and for their ongoing efforts to address the main issues.
> >
> > Most of my concerns were resolved by the authors' feedback, and I remain positive about the overall contribution of this work.
> >
> > Despite these significant improvements, I would like to raise a minor concern regarding the use of GAN-based methodologies in medical imaging. In clinical and medical contexts, GAN-generated images may sometimes introduce subtle artifacts or unrealistic features, which could affect the interpretability and reliability of the simulation results. I would encourage the authors to briefly discuss these limitations and the need for careful validation when using GAN-based simulation in the medical domain, perhaps in the discussion or limitations section.
> >
> > Regarding this issue, I'm positive to increase my rating.
> >
> > Overall, the manuscript has been substantially improved, and I look forward to its contribution to NeurIPS.

---

> > > ### Author Response · Authors · 2025-08-06
> > > **Further Discussion on GAN-based approaches**
> > >
> > > We sincerely thank the reviewer for the constructive feedback. We adopt GAN in our work since it is a well-established technique in medical imaging. Existing approaches demonstrate promising results in generating different diagnostic modalities, such as MRI, X-ray, or ultrasound [1, 2, 3]. However, we agree that GAN-based approaches can introduce artifacts or unrealistic features, which may limit their applicability in clinical settings that demand high structural fidelity. Accordingly, in the **Conclusion—Limitations and future work** section of the original manuscript, we acknowledged that our current ultrasound simulation lacks physics-based constraints. We also noted that this motivates our future work toward investigating novel methods to improve the realism and diversity of ultrasound simulations.
> > >
> > > Specifically, in the context of orthopedic surgery, we are aware that the high-intensity bone surfaces (important for clinical application) predicted by GAN models may slightly deviate from the true anatomical structures. Additionally, when generating 2D medical images from different viewing angles, inconsistencies can arise in overlapping regions of the same anatomy. To address these issues, promising future directions include incorporating physics-informed losses or novel network architectures to improve both the anatomical accuracy and the cross-view consistency of the predicted ultrasound images. To evaluate such new approaches, particular metrics that assess bone surface fidelity and spatial consistency should also be incorporated.
> > >
> > > We also believe that our open-source platform can serve as a foundation for the community to explore and mitigate these limitations. Although we are currently unable to update the submitted manuscript, we will incorporate a more detailed discussion on the limitations of GAN-based approaches in orthopedic applications in a future version.
> > >
> > > [1] Singh, Nripendra Kumar, and Khalid Raza. "Medical image generation using generative adversarial networks: A review." Health informatics: A computational perspective in healthcare (2021): 77-96.
> > >
> > > [2] Kora Venu, Sagar, and Sridhar Ravula. "Evaluation of deep convolutional generative adversarial networks for data augmentation of chest x-ray images." Future Internet 13.1 (2020): 8.
> > >
> > > [3] Song, Yuhan, and Nak Young Chong. "S-cyclegan: Semantic segmentation enhanced ct-ultrasound image-to-image translation for robotic ultrasonography." 2024 IEEE International Conference on Cyborg and Bionic Systems (CBS). IEEE, 2024.

---

> > > > ### Comment · Reviewer_nCZp · 2025-08-06
> > > > **Further discussion**
> > > >
> > > > I would like to express my sincere appreciation for providing further discussion regarding the limitations of GAN-based approaches and possible future directions.
> > > >
> > > > I believe the authors’ responses adequately address my concerns. Additionally, I also believe that the authors' clarification of current limitations and their commitment to exploring physics-informed and improved GAN-based methods strongly demonstrate a clear direction for advancing research in the medical field.
> > > >
> > > > I am satisfied with the authors’ response, and I will increase my rating from 5 to 6. Further, I hope that this work will be accepted at the NeurIPS conference.
> > > >
> > > > Thank you for the authors' effort in the rebuttal phase.

---

### Decision · Program_Chairs · 2025-09-18

**Decision:**

Accept (poster)

**Comment:**

This paper presents SonoGym, a scalable and high-performance simulation platform specifically designed for robotic ultrasound tasks in surgical settings. All reviewers reach concensus that the manuscript is ready for publication. Pleae revise the paper based on reviewers comment.